# Novel Isoniazid-Carborane Hybrids Active In Vitro against *Mycobacterium tuberculosis*

**DOI:** 10.3390/ph13120465

**Published:** 2020-12-15

**Authors:** Daria Różycka, Małgorzata Korycka-Machała, Anna Żaczek, Jarosław Dziadek, Dorota Gurda, Marta Orlicka-Płocka, Eliza Wyszko, Katarzyna Biniek-Antosiak, Wojciech Rypniewski, Agnieszka B. Olejniczak

**Affiliations:** 1Institute of Medical Biology, Polish Academy of Sciences, 106 Lodowa St., 93-232 Lodz, Poland; drozycka@cbm.pan.pl (D.R.); mkorycka@cbm.pan.pl (M.K.-M.); jdziadek@cbm.pan.pl (J.D.); 2Institute of Medical Sciences, Medical College, University of Rzeszow, 2A Kopisto Avenue, 35-959 Rzeszow, Poland; ania.zaczek@yahoo.pl; 3Institute of Bioorganic Chemistry, Polish Academy of Sciences, 12/14Z. Noskowskiego St., 61-704 Poznan, Poland; d_gurda@ibch.poznan.pl (D.G.); mplocka@ibch.poznan.pl (M.O.-P.); wyszkoe@ibch.poznan.pl (E.W.); kbiniek@ibch.poznan.pl (K.B.-A.); wojtekr@ibch.poznan.pl (W.R.)

**Keywords:** boron cluster, carborane, isoniazid, antitubercular activity, *Mycobacterium tuberculosis*

## Abstract

Tuberculosis (TB) is a severe infectious disease with high mortality and morbidity. The emergence of drug-resistant TB has increased the challenge to eliminate this disease. Isoniazid (INH) remains the key and effective component in the therapeutic regimen recommended by World Health Organization (WHO). A series of isoniazid-carborane derivatives containing 1,2-dicarba-*closo*-dodecaborane, 1,7-dicarba-*closo*-dodecaborane, 1,12-dicarba-*closo*-dodecaborane, or 7,8-dicarba-*nido*-undecaborate anion were synthesized for the first time. The compounds were tested *in vitro* against the *Mycobacterium tuberculosis* (*Mtb*) H37Rv strain and its mutant (Δ*katG*) defective in the synthesis of catalase-peroxidase (KatG). *N*′-((7,8-dicarba-*nido*-undecaboranyl)methylidene)isonicotinohydrazide (**16**) showed the highest activity against the wild-type *Mtb* strain. All hybrids could inhibit the growth of the Δ*katG* mutant in lower concentrations than INH. *N*′-([(1,12-dicarba-*closo*-dodecaboran-1yl)ethyl)isonicotinohydrazide (**25**) exhibited more than 60-fold increase in activity against *Mtb* Δ*katG* as compared to INH. This compound was also found to be noncytotoxic up to a concentration four times higher than the minimum inhibitory concentration 99% (MIC_99_) value.

## 1. Introduction

According to the World Health Organization (WHO), tuberculosis (TB) causes the highest number of deaths due to infectious disease worldwide. TB is a curable disease that requires long-term treatment with multiple drugs. Chemotherapy for drug-susceptible TB includes the four first-line TB drugs, namely isoniazid (INH), rifampicin (RIF), pyrazinamide (PZA), and ethambutol (EMB), for the first two months of treatment, followed by INH and RIF for the next four months of treatment [1].

New effective drugs are necessary to reduce the duration of TB treatment, as well as for the treatment of multidrug-resistant TB (MDR; defined as caused by *Mycobacterium tuberculosis* (*Mtb*) resistant to at least RIF and INH), extensively drug-resistant TB (XDR; defined as MDR and additional resistance to at least one fluoroquinolone and one second-line injectable drug), and totally drug-resistant TB (TDR) [2,3,4]. 

Prodrug INH, an isonicotinic acid-derivative hydrazide (pyridine-4-carbohydrazide), is activated in bacilli by catalase-peroxidase (KatG) to form the INH-nicotinamide adenine dinucleotide (NAD) adduct. This adduct inhibits the enzyme enoyl-acyl carrier protein reductase (InhA) of the fatty acid synthase II (FASII), leading to the growth inhibition or death of bacilli. INH is a first-line drug recommended by WHO in the treatment of drug-sensitive tuberculosis [5]. The accumulation of mutations in *katG*, as well as in the promoter region of *inhA*, are the primary mechanisms of resistance to INH responsible for about 75% of all causes of *Mtb* resistance to INH in clinical settings [6,7]. The INH-resistant strains carrying mutations in the *ndh*, *kasA*, or *oxyR*-*ahpC* intergenic regions were also reported, but their roles in the resistance remain unclear [8]. INH is a prodrug, and its chemical modifications in its core structure could improve its bioavailability or membrane permeability [9,10,11]. The permeation of INH throughout the bacterial cell envelope and its activity were improved when lipophilic moieties were introduced into the framework of INH [10]. INH derivatives presenting increased lipophilicity could compose the potent bactericidal compound effective against tubercle bacilli [12]. 

Boron clusters are polyhedral caged compounds [13]. The most famous is icosahedral dicarba-*closo*-dodecaborane (carborane, C_2_B_10_H_12_) and its three isomeric forms: 1,2-C_2_B_10_H_12_ (*ortho*-), 1,7-C_2_B_10_H_12_ (*meta*-), and 1,12-C_2_B_10_H_12_ (*para*-), depending on the position of the carbon atoms within the carborane structure [13]. The biomedical application of carboranes has been discussed [14,15,16,17,18]. The properties of carborane clusters that can be used in medical chemistry include the following: inorganic nature and resistance to enzymatic degradation, susceptibility to orthogonal functionalization, the possibility of interactions with peptides, and tendency to self-assemble in an aqueous solution [18].

One of the important features that can distinguish carboranes from other molecules is lipophilicity. The lipophilicity changes depending on the carborane isomer in the following order: *ortho*-carborane < *meta*-carborane < *para*-carborane. The presence of a partial negative charge located on boron-bound hydrogen atoms in BH groups, their “hydride-like” characteristics, and the inability to form a classical hydrogen bond have an impact on their lipophilic character [14,19]. 7,8-Dicarba-*nido*-undecaborate anion (*nido*-carborane, 7,8-C_2_B_9_H_11_(–1)), with amphiphilic properties, is obtained from *ortho*-carborane (Figure 1) [13].

Herein, we propose the use of dicarba-*closo*-dodecaboranes (*ortho*-, *meta*-, and *para*-carborane) and 7,8-dicarba-*nido*-undecaborate anion (*nido*-carborane) as leads for the design and synthesis of novel INH analog and INH hybrids in order to assess the effect of carborane cluster and its properties on the in vitro antibacterial activity of modified INH against the *Mtb* H37Rv strain and its mutant strain defective in the synthesis of a functional KatG (Δ*katG*).

## 2. Results and Discussion 

### 2.1. Chemistry

#### 2.1.1. Synthesis of Isoniazid-Carborane Cluster Conjugates

The newly synthesized INH analog containing *para*-carborane cluster **3** (Figure 2), instead of a pyridine ring, was obtained in a simple three-step procedure: (1) the synthesis of 1,12-dicarba-*closo*-dodecaborane-1-carboxylic acid (**1**) [20], (2) the synthesis of 1,12-dicarba-*closo*-dodecaborane-1-carboxylic acid chloride (**2**) [20], and (3) the synthesis of 1,12-dicarba-*closo*-dodecaborane-1-carboxylic acid hydrazide (**3**). This synthesis was performed in one glass flask, and the yield of product **3** after isolation and purification by column chromatography was 41%.

INH hybrids **8**–**10**, **14**–**16**, and **20**–**25** were prepared by the functionalization of INH at *N*-2 (Figure 3). Treatment of INH (**4**) with the appropriate *N*-succinimidyl active esters **5**–**7** containing *ortho*-/*meta*-/*para*-carborane clusters [21] for three to four days at room temperature (RT) in absolute EtOH typically led to the condensation products hydrazide **8**–**10** without complication. The yield of products **8**–**10** after isolation and purification by column chromatography varied between 56% and 72%, depending on the type of the carborane cluster, with a lower yield for the 1,2-dicarba-*closo*-dodecaborane-bearing derivative **8**.

Reductive amination of INH (**4**) with an appropriate aldehyde-bearing *ortho*-/*meta*-/*para*-carborane group **11**–**13** or **17**–**19** [22] in absolute EtOH (for **14** and **15** and **20**–**22**) or anhydrous ethyl acetate (for **16**) at 35–40 °C or at RT (for **18**) led to the corresponding isonicotinoyl hydrazones 14–16 and 20–22, which were generally isolated as a crystalline solid after isolation and purification by column chromatography. The yields varied 50–84% (for **14**–**16**) and 55–79% (for **20**–**22**).

It should be noted that the synthesis of conjugate **16** was performed using aldehyde **11** bearing the *closo* form of the carborane cluster. During the synthesis, we observed a slow transformation of the electroneutral *closo*-carborane into a negatively charged *nido*-cage, resulting in the formation of compound **16** with *nido*-carborane, with approximately 20% yield. After separation and purification by column chromatography, the product containing a closed form of the carborane cluster was transformed into *nido*-carborane. We observed that the product with *closo*-carborane stored in the refrigerator for 10 days transformed into compound **16** with 50% yield. The same transformation was observed at RT, without solvent and with solvent, which led to the formation of **16** with 65% yield. The formation of the *nido*-form of the carborane cluster during the synthesis of compounds **8**, **20**, and **23** was not observed. 

Hawthorne described the synthesis of *nido*-monoanion 7,8-C_2_B_9_H_12_(–1) by the alcoholic base degradation of *ortho*-carborane and substituted derivatives [23]. Tertiary amines, hydrazine, ammonia, piperidine, pyrrolidine, and fluoride ions have also been used to obtain *nido*-carborane [24]. Removal of the BH vertex occurs regiospecifically at the most electropositive BH vertex, which is B(3) or B(6) in 1,2-C_2_B_10_H_11_.

It was found that the *closo*-carborane derivatives can be cleaved to *nido*-structure in solution. The spontaneous degradation was observed for racemic *ortho*-carboranylalanine and *ortho*-carboranyl-lactose conjugates in a water–methanol solution [25] and in water or methanol [26], respectively. 1,2-Bis(aminomethyl)-1,2-dicarba-*closo*-dodecaborane hydrochloride was also converted to an opened form in deuterated DMSO [27].

The stereochemistry of the double bond in the hydrazones **14**–**16** and **20**–**22** was assigned as synperiplanar E on the basis of ^1^H NMR experiments. Hydrazone derivatives containing an acyl group may exist as E/Z geometric isomers with C=N double bonds as synperiplanar or antiperiplanar amide conformers. However, it is reported that hydrazones obtained using aldehydes and substituted hydrazides are present in the solution as E isomers [28]. HPLC analysis of our carborane–INH hybrids confirmed that only one isomer was present in all hybrids. NOESY experiments were conducted in DMSO-d_6_ and revealed a well-defined cross peak between the CO-NH proton and the iminic proton; it is only possible in E geometry. Another cross-peak between the CO-NH proton and the 2a and 2a′ protons revealed synperiplanar amide conformers.

We attempted to reduce the double bond by using NaBH_3_CN/NaBH_4_ in compounds **14**–**16**. Unfortunately, we did not observe the expected product; the unreacted substrate remained in the reaction mixture. Hydrazides **23**–**25** were obtained by reduction of the parent compounds **20**–**22** with freshly recrystallized NaBH_3_CN [29] in anhydrous EtOH and further purification by column chromatography (yield 53–80%). The direct synthesis of **23**–**25** without the isolation of **20**–**22** was unsuccessful and was therefore abandoned.

The products **3**, **8**–**10**, **14**–**16**, and **20**–**25** were characterized by ^1^H, ^13^C, ^11^B NMR, FTIR, MS, RP-HPLC (Appendix A (SM)), and TLC.

#### 2.1.2. X-Ray Structure Analysis

Each crystal structure contains in the asymmetric unit one molecule of INH–carborane conjugates **14**, **15**, and **21**. The asymmetric unit of **21** contains in addition a molecule of methanol (Figure 4). A comparison of the three crystal structures reveals similarities in the interactions of the molecules. In each crystal structure, there are interactions between the N atoms of the pyridine ring and the C-H groups of the carborane cluster of the neighboring molecules. Apparently, the C-H bond is polarized sufficiently to form weak hydrogen bonds (Table 1). 

In addition, regular hydrogen bonds are formed in **14** and **15** between the NH groups of the INH residues and the carbonyl oxygen atoms of the neighboring molecules, thus chain-linking the molecules. In **21**, the chains have interposing hydroxyl groups of MeOH molecules. The above two types of interactions form the crystal lattice in the three structures (Figure 5).

The crystal structures of the INH-carborane derivatives **14**, **15**, and **21** demonstrate that the hydrogen bonding potential of the compounds is not limited to the H-bond donors or acceptors present on the INH residue. 

The carborane clusters can also enter into weak H-bonding interactions through the C-H group. In this respect, the position of the C atom within the cluster is significant. The acidic nature of the C-H group was previously observed for free carboranes [30].

The interactions described could be relevant in the ligand–receptor complex formation. It was found that 1-carba-*closo*-dodecaborane (CB_11_H_12_)^–^, a model for metallacarborane inhibitors of HIV protease, interacts with the building blocks (amino acids) of biomolecules by the formation of dihydrogen bonds, using B-H groups of the carborane cluster. Dihydrogen bonds are mainly electrostatic interactions between negatively charged boron-bound hydrogen atoms and positively charged hydrogen atoms of biomolecules. Another type of interaction was found for C-H ··Y hydrogen-bonded complexes. These complexes were less stable [31,32].

### 2.2. Biological Investigation

#### 2.2.1. Antimycobacterial Activity of Isoniazid-Carborane Cluster Conjugates

The mycobactericidal activity of the obtained compounds was examined. All the tested compounds were evaluated in vitro against the wild-type *Mtb* strain and its mutant carrying *katG* with inner deletion (Δ*katG*) exclusively and unable to synthesize a functional KatG. A clinically used drug INH (**4**) was included in the assay as a reference compound. 

As expected, compound **3** was not sufficiently active; the growth of *Mtb* was not considerably inhibited when 75 µg/mL (0.37 mM) of this compound was used. Therefore, further studies of this conjugate were discontinued. 

Hydrazide–hydrazone derivatives are present in many bioactive molecules and show a wide variety of biological activities such as antibacterial [33], antitubercular [33], antifungal [33], anticancer [34] anti-inflammatory [35], anticonvulsant [36], and antiviral [37,38] activities. 

INH hydrazide–hydrazone derivatives are known to show promising anti-TB properties. These compounds include monosubstituted-benzylidene INH derivatives. They exhibited a significant activity against *Mtb* (minimum inhibitory concentration (MIC) 0.31 µg/mL) when compared with INH (**4**) (MIC 0.20 µg/mL) and, thus, could be a promising starting point to developing new lead compounds [10].

The test results for compounds **8**–**10**, **14**–**16**, and **20**–**25** are presented in Table 2 in terms of the minimum inhibitory concentration resulting in 50% and 99% inhibition (MIC_50_ and MIC_99_, respectively) of *Mtb* growth. Hydrazides **8**–**10** did not significantly affect the growth of *Mtb*. Their MIC_50_ (22.4–44.7 µM) and MIC_99_ (74.5–100 µM) values were higher than the MIC_50_ (0.073 µM) and MIC_99_ (0.36 µM) values of INH (**4**).

The other tested conjugates exhibited significant activity against *Mtb*. Hydrazone **16** bearing amphiphilic 7,8-dicarba-*nido*-undecaborate anion showed the highest antimycobacterial activity (MIC_99_ 0.33 µM). Its MIC_99_ value is comparable to that of INH (**4**).

The *nido*-carboranes have not been studied as often as *closo*-carboranes. It was shown that the indomethacin-*nido*-carborane conjugate revealed higher water solubility and improved stability compared to the indomethacin-*ortho*-carborane conjugate. It was also observed that the presence of *nido*-carborane affected increased inhibitory potency and selectivity for COX-2 with respect to the respective phenyl analog [39].

For today nucleic bases–carborane conjugates were tested for their antitubercular activity [40,41]. Thymine derivatives containing *ortho*-, *para*-carborane, or *nido*-carborane showed different activities against *Mtb* and *M*. *smegmatis*, depending on the form of carborane cluster [41]. *Closo*-carborane and *nido*-carborane exhibited similarities and, also, differences—single negative-charge (*nido*-form) or no charge (*closo*-form)—used in the present study. The negative charges are distributed on all cluster atoms and the attached H atoms. These differences affect their physicochemical and other properties influencing their biological activity. Indeed, though the type of boron cluster and its *closo*/*nido* status may affect the biological activities of their derivatives, the cause for this phenomenon is not clear at present and requires further study.

Schiff base **21** modified with *meta*-carborane was slightly less active (MIC_99_ 0.82 µM) than **16**. Conjugates **14**, **20**, and **22**–**25** exhibited significant activity against *Mtb* (MIC_99_ 1.6–6.5 µM) but showed lower activity than compounds **16** and **21**. Schiff bases **21** and **22** with a longer linker between the carborane and INH residue were more active than the appropriate Schiff bases **14** and **15** with a shorter linker. 

The different antimycobacterial activity of compounds, **8**–**10**, **14**–**16**, and **20**–**25** is most probably related to the steric characteristic of the carborane cluster and its impact on the promotion of the formation of the isonicotinoyl radical to form INH–nicotinamide adenine (NAD) adducts.

It is previously reported that access to the heme active site of KatG poses steric constraints, and therefore, any factor influencing the stereochemistry of tested compounds should be relevant [42]. The manganese catalyst (Mni^v^-Mni^v^(µ-O)_3_L_2_)(PF_6_)_2_, (L=1,4,7-trimethyl-1,4,7-triazacyclononane) was used, as it was hypothesized to be a mimic for oxidation by the KatG enzyme due to its catalase activity [43]. This catalyst forms a Mn^V^=O species due to its catalase activity [44,45]. When INH was reacted with various oxidants, including a manganese catalyst, in the presence of the stable radical trap 2,2,6,6-tetramethylpiperidine-1-oxyl (TEMPO), *O*-isonicotinoyl hydroxylamine was formed [44,45]. This product is the consequence of acyl radical generation and trapping, as supported by Braslau [46]. 

In our investigation, INH (**4**) was reacted in the presence of TEMPO and the Mn catalyst under nitrogen atmosphere in MeOH/ACN (1:99, *v*/*v*) with periodic acid as a co-oxidant [47,48]. *O*-Isonicotinoyl hydroxylamine (**26**) was formed (Appendix A, general experimental details, (SM)), and an unreacted substrate INH (**4**) was not observed in the reaction mixture. When elected hybrids **8**, **21**, and **25**, with diverse antimycobacterial activity against *Mtb* (Table 2), were reacted with a manganese-containing oxidant in the presence of TEMPO, *O*-isonicotinoyl hydroxylamine (*26*) was produced. The unreacted substrates were also observed in the reaction mixture (3–46%) for the lowest value for the most active hybrid **21** of these three compounds (Appendix A (SM)). These results could shed light on the different antimycobacterial of the tested conjugates compared to unmodified INH (**4**). It was also suggested that the formation of the INH analog adducts with NAD requires a sufficiently long-lived acyl radical and that, for aromatic hydrazides, the nature of substituents play an important role in the stabilization of radical intermediates involved in the overall process of INH activation [49].

A comparison between the hydrazones **20**–**22** and the corresponding hydrazides **23**–**25** shows that the hydrazones are more active (MIC_99_ 0.82–1.6 µM) than their analogs (MIC_99_ 3.35–6.5 µM), as already reported by some authors [9], possibly due to their hydrolysis and the subsequent production of the acyl radical as a result of INH (**4**) activation by KatG.

Given the results obtained for the *Mtb* strain, we conducted the testing of INH–carborane hybrids against the mutant defective in the synthesis of a functional KatG (Δ*katG*). The MIC_99_ values showed that all hybrids containing the carborane cluster are more effective than INH (**4**) against Δ*katG* (Table 2). The MIC_99_ values of compounds **8**–**10**, **14**–**16**, and **20**–**24** were two to nine times lower than that of INH (**4**). Hydrazide **25**, modified with the *para*-carborane cluster, is the most potent derivative (MIC_99_ 24.4 µM) and shows a 61-fold increase in activity as compared to INH (**4**) (MIC_99_ 1500 µM). Taking together, its potent activity against the wild-type *Mtb* strain (MIC_50_ 0.32 µM) and the INH-resistant mutant (Δ*katG*, MIC_50_ 3.25 µM), compound **25** seems to be promising with regards to the development of its new derivatives library. Interestingly, conjugate **16** containing *nido*-carborane was very active against *Mtb* but not so active against the Δ*katG* strain (MIC_99_ > 660 µM). 

The formation of a covalent adduct with the NAD cofactor and inhibition of FASII might be not the only mechanism of action of the INH-modified compounds. Hydrazone **20** and hydrazides **23**–**25** are active against wild-type *Mtb*, but their MIC_99_ are one order of magnitude higher compared to INH (**4**) and **16**. On the other hand, they are very active (especially conjugate **25**) against mutant Δ*katG*. Therefore, at least compound **25** does not require the formation of an adduct with NAD to present the bacteriostatic/bactericidal effect. We cannot exclude that compound **25** can form the adduct within the wild-type strain when KatG is available and that INH derivatives synthesized here, with or without a NAD cofactor, affect the same molecular target as INH. However, it is also likely that other essential molecules of *Mtb* are targeted by this compound. We speculate that compounds that are effective against wild-type *Mtb* but not the Δ*katG* mutant (compound **16**) need activation by the formation of a covalent adduct with the NAD cofactor and affect InhA; however, the compounds that are potent against Δ*katG* do not need the intracellular activation and target other essential molecules within *Mtb*. We were not able to exclude that some compounds, such as **25**, can compose an INH–NAD adduct in the wild-type *Mtb*, but it presents the bactericidal effect without a NAD cofactor as well, affecting the same or other targets. The identification of the alternative targets for INH derivatives synthetized in this work needs complex study. Research of direct InhA inhibitors, avoiding the activation step, has emerged as a promising strategy to combat the global spread of MDR-TB [50].

The obtained results seem to suggest that the modification of INH (**4**) to improve its activity may indeed be a feasible approach to overcome resistance in *katG*. 

#### 2.2.2. In Vitro Cytotoxicity Assay 

To determine whether the obtained isoniazid-carborane hybrids that demonstrate antimycobacterial properties as confirmed by their MIC_50_ and MIC_99_ values are promising lead compounds, we performed cellular cytotoxicity analyses. We chose normal human keratinocytes (HaCaT) for this study. The cell viability was estimated using the (3-(4,5-dimethylthiazol-2-yl)-2,5-diphenyltetrazolium bromide (MTT) tetrazolium salt assay. The cells were incubated with the compounds for 24 h with the concentration that corresponds to the previously estimated MIC_50_ and MIC_99_ values for the wild-type *Mtb* and Δ*katG* strains. To assess whether the solvent (DMSO) itself affects the viability of the cells, we incubated the cells with DMSO in the amount corresponding to the highest concentration of the tested compound. 

The results indicate that the analyzed compounds applied in concentrations estimated for the wild-type strain did not affect the cell viability (>80% of live cells), except for compound **8**, where only 50% of the cells were viable after 24-h treatment at a concentration consistent with the MIC_99_ value (Figure 6A). The analyzed compounds tested at concentrations that were estimated as MIC_50_ and MIC_99_ values for the Δ*katG* strain were more toxic to HaCaT cells (Figure 6B). Two of them (compounds **9** and **25**) did not cause severe cytotoxicity, but only **25**, showing the highest activity against Δ*katG*, did not affect cell viability at both the MIC_50_ and MIC_99_ values. 

Additionally, the cytotoxicity of compounds **8**–**10**, **14**–**16**, and **20**–**25** was established by measuring the 50% cytotoxic concentration (CC_50_) (Table 2). The selectivity index (SI) was determined as the ratio of the measured CC_50_ to the MIC_99_ of *Mtb* or to the MIC_99_ of Δ*katG* (Table 2). Nearly all the tested hydrazones (**14**–**16** and **21**–**22**) and hydrazides (**23**–**25**) exhibited potent anti-*Mtb* activity combined with low cytotoxicity that resulted in SI values ranging from 756 to 15, with the highest value for the most active conjugate **16**. Hydrazide **25** showed a CC_50_ value of 98.45 µM, which is four times higher than its MIC_99_ value against the mutant strain. 

#### 2.2.3. Apoptosis/Necrosis Assay by Flow Cytometry

A detailed analysis of cell toxicity induction was performed by the flow cytometry apoptosis/necrosis assay. Dual staining with YO-PRO1 and propidium iodide (PI) fluorochromes was analyzed using a FACSCalibur flow cytometer after 24 h of treatment (Figure 7). The obtained results showed that the analyzed compounds are much less toxic to HaCaT cells at concentrations estimated as MIC_50_ and MIC_99_ values for the *Mtb* strain (Figure 7A). We found only a small percentage (up to 14%) of apoptotic/necrotic cells in almost all analyzed samples. Compound **8** alone decreased the cell viability, resulting in 16.45% of apoptotic/necrotic cells after 24-h treatment. The MIC_50_ and MIC_99_ values calculated for the Δ*katG* strain strongly affected the cell viability. After 24 h of treatment, a high percentage of apoptotic/necrotic cells was observed (Figure 7B). Compound **25** alone did not affect the cellular toxicity, even at the concentration corresponding to the MIC_99_ value being the most promising among the tested compounds. A 24-h, the incubation with compound **25** led to a low number of apoptotic/necrotic cells at both MIC_50_ and MIC_99_ concentrations (Figure 7C,D). Concentrations corresponding to the MIC_50_ and MIC_99_ values for the *Mtb* strain resulted in approximately 87% viability, whereas the same values estimated for the Δ*katG* strain resulted in cell viability of 87.45% and 82.20%, respectively. 

### 2.3. Physicochemical Investigation

#### 2.3.1. Lipophilicity Measurement

Lipophilicity of a compound is associated with many physicochemical and physiological properties. Lipophilicity influences the solubility, permeability, oral absorption, cell uptake, blood-brain penetration, and metabolism [51].

The lipophilicity of the synthesized compounds **8**–**10**, **14**–**16**, and **20**–**25** was measured as the partition (*P*) or distribution (*D*) coefficient (Table 3). The octan-1-ol/water partition/distribution coefficient is a widely used parameter for measuring lipophilicity. Octan-1-ol has a hydroxyl group, which is the donor and acceptor of hydrogen bonds, and a hydrocarbon chain, which, to some extent, allows the mimic of natural membrane barriers. The log *P* value is the ratio of the concentration of the neutral compound in water and octan-1-ol. Log *D* is the log partition at a particular pH and is reserved to compound partially or completely ionized in the aqueous phase. [52]. There are various computational methods to predict the log *P* and log *D*. A potential drawback of using the calculated lipophilicity is that the methods of calculation have systemic errors [53].

The presence of a carborane moiety in a modified isoniazide structure increases its lipophilicity as compared to INH (**4**). The lipophilicity of the modified compounds was from one to three orders of magnitude higher than that of the unmodified counterpart. For comparison, the mother compound INH (**4**) showed a log *P* value of –1.00 ± 0.09 (hydrophilic compound). The log *P* values for compounds **14** and **15** were 2.03 ± 0.09 and 2.28 ± 0.20, respectively. These are the highest values of log *P* among compounds **8**–**10** and **20**–**25** modified with *closo*-carborane. 

The removal of the most electrophilic boron atom in lipophilic, neutral *closo*-carborane resulted in the formation of the more hydrophilic, anionic *nido*-carborane [18]. The log *D*_7.4_ value for **16** was 0.67 ± 0.04 and was lower than log *P* values of conjugates **14** and **15** containing the *closo*- form of the carborane cluster, as expected, but still one order of magnitude higher than the log *P* value of the unmodified counterpart. 

#### 2.3.2. Parallel Artificial Membrane Permeability Measurement

For the early prediction of absorption, parallel artificial membrane permeability (PAMPA) is one of the most frequently used in vitro models. PAMPA is a simple and robust method to forecast the transcellular passive absorption through membranes [54]. Briefly, the technique involves two well plates forming a “sandwich”-like structure—a donor plate with porous membrane wells that are submerged in the wells of an acceptor plate. The porous membrane is coated with a lipid solution to form a bilayer to mimic the biological membrane. The donor wells contain the drug in an aqueous buffer solution, and the acceptor wells contain only a buffer solution. Drugs that can permeate the cell membrane will passively diffuse into the acceptor wells. By changing the lipid composition, different membranes can be simulated. After the incubation time, the ratio of a compound that crossed the artificial membrane is calculated and used to obtain the effective passive permeability value *P*_e_ (cm/s).

To predict the passive membrane permeability, the most critical parameter is lipophilicity [55]. 

The PAMPA assay was conducted to evaluate the permeability of compounds **4**, **8**–**10**, **14**–**16**, and **20**–**25** through an artificial membrane. The artificial membrane was composed of 2% egg lecithin in *n*-dodecane. Each compound was placed on the donor side of the membrane. After 18 h of incubation at RT, the amount of the compound was quantified through UV analyses. The effective permeability coefficient (*P*_e_) was calculated as described in Materials and Methods. The PAMPA results indicated that the INH derivative containing the carborane clusters **8**–**10**, **14**–**16**, and **20**–**25** exhibited better permeation than INH (**4**) (Table 3). INH (**4**) was an impermeable compound, while INH–carborane hybrids showed high membrane permeability. The permeation of the modified compounds **9**, **16**, and **20** was one order of magnitude higher than that of the unmodified counterpart. For other modified hybrids, the permeation was two orders of magnitude higher (**8**, **10**, **14**, and **21**–**25**), three orders of magnitude higher (**15**) than that of INH (**4**). These modified INH–carborane hybrids showed better lipophilic characteristics, as indicated by their log *P*/*D*, and better membrane permeability, as indicated by their log *P*_e_, as compared to INH (**4**), but there was no clear relationship between the lipophilicity/permeation and in vitro activity against the *Mtb* and Δ*katG* strains.

## 3. Materials and Methods

### 3.1. Chemistry

Most of the chemicals were obtained from the Alfa Aesar (Haverhill, MA, USA) and were used without further purification unless otherwise stated. Lecithin (l-α-phosphatidylcholine from egg yolk, type XVI-E, ≥99% (TLC)) and dodecane were purchased from Sigma-Aldrich (Steinheim, Germany). Flash chromatography was performed using silica gel 60 (230–400 mesh, ASTM, Aldrich Chemical Company). *R*_f_ values referred to analytical TLC performed using precoated silica gel 60 F254 plates purchased from Sigma-Aldrich (Steinheim, Germany) and developed in the indicated solvent system. Carborane was purchased from KATCHEM spol. s r.o. (Reź/Prague, Czech Republic). Compounds were visualized using UV light (254 nm) and 0.5% acidic solution of PdCl_2_ in HCl/methanol for boron-containing derivatives. The yields were not optimized. ^1^H NMR, ^13^C NMR, and ^11^B NMR spectra were recorded on a Bruker Avance III 600 MHz spectrometer equipped with a direct ATM probe. The spectra for ^1^H, ^13^C, and ^11^B nuclei were recorded at 600.26 MHz, 150.94 MHz, and 192.59 MHz, respectively. Deuterated solvents were used as standards. For NMR, the following solvents were used: C_5_D_5_N (*δ*_H_ = 7.19, 7.55, 8.71, *δ*_C_ = 123.50, 1135.50, 149.50 ppm) and CD_3_OD (*δ*_H_ = 3.35, *δ*_C_ = 50.00 ppm). All chemical shifts (*δ*) are quoted in parts per million (ppm) relative to the external standards. The following abbreviations are used to denote the multiplicities: s = singlet, d = doublet, dd = doublet of doublets, ddd = doublet of doublets of doublets, t = triplet, dt = doublet of triplets, q = quartet, quin = quintet, bs = broad singlet, and m = multiplet. *J* values are expressed in Hz. Mass spectra were performed on a PurIon S (Teledyne ISCO, Lincoln, NE, USA). For compound **16**, the ionization was achieved by electrospray ionization in the negative ion mode (ESI–). The capillary voltage was set to 2.5 kV. The source temperature was 200 °C, and the desolvation temperature was 350 °C. Nitrogen was used as a desolvation gas (35 L/min, purity >99%, nitrogen generator EURUS35 LCMS, E-DGSi SAS, Evry, France). For compounds **3**, **8**–**10**, **14**, **15**, and **20**–**25**, the ionization was also achieved by atmospheric pressure chemical ionization (APCI). The entire flow was directed to the APCI ion source operating in the positive ion mode. Total ion chromatograms were recorded in the *m*/*z* range of 100 to 600. The vaporization and capillary temperatures were set at 250–400 and 200–300 °C, respectively. Capillary voltage was set at 150 V and corona discharge at 10 µA. The theoretical molecular masses of the compounds were calculated using the “Show Analysis Window” option in the ChemDraw Ultra 12.0 program. The calculated *m*/*z* corresponded to the average mass of the compounds consisting of natural isotopes. Infrared absorption spectra (IR) were recorded using a Nicolet 6700 Fourier-transform infrared spectrometer from Thermo Scientific (Waltham, MS, USA) equipped with an ETC EverGlo* source for the IR range, a Ge-on-KBr beam splitter, and a DLaTGS/KBr detector with a smart orbit sampling compartment and diamond window. The samples were placed directly on the diamond crystal, and pressure was induced to make the surface of the sample conform to the surface of the diamond crystal. UV measurements were performed using a GBC Cintra10 UV-Vis spectrometer (Dandenong, Australia). The samples used for the UV experiments, ca. 0.5 A_260_ optical density units (ODUs) of each compound, were dissolved in CH_3_OH. The measurement was performed at RT.

Partition coefficient measurements and PAMPA were performed using a Thermo Scientific^TM^ Varioskan^TM^ Flash Multimode Reader equipped with UV-Star^®^ 96-well plates (Greiner Bio-One GmbH, Frickenhausen, Germany). RP-HPLC analysis was performed on a Hewlett-Packard 1050 system equipped with a UV detector and a Hypersil Gold C18 column (4.6 × 250 mm, 5 µm particle size, Thermo Scientific, Runcorn, UK). UV detection was conducted at λ = 262 nm. The flow rate was 1 mL min^–1^. All analyses were run at ambient temperatures. The gradient elution was as follows: gradient A—20 min from 0% to 100% B, 20 min at 100% B, and 15 min from 100% to 0% B. Buffer A contained 0.1-M TEAB (triethylammonium bicarbonate), pH 7.0, in acetonitrile:water (2:98), and buffer B contained 0.1-M TEAB, pH 7.0, in acetonitrile:water (40:60). Gradient B—20 min from 0% to 100% B, 20 min at 100% B, and 15 min from 100% to 0% B. Buffer A contained 0.1-M TEAB, pH 7.0, in acetonitrile:water (2:98), and buffer B contained 0.1-M TEAB, pH 7.0, in acetonitrile:water (60:40).

Crystals of **14**, **15**, and **21** were obtained by slow evaporation from ethanol with a few drops of water. X-ray diffraction measurements on **14** and **15** crystals were performed under cryogenic conditions on a Rigaku Oxford Diffraction Xcalibur four-circle diffractometer (Yarnton, Oxfordshire, England) equipped with a Mo-sealed tube anode and an EosS2 CCD detector, while measurements on **21** were performed on a SuperNova (Agilent) diffractometer equipped with a Cu-sealed tube lamp. The data were processed using CrysAlisPro software from Rigaku Oxford Diffraction, and the structures were solved and refined using SHELXT and SHELXL programs through the Olex^2^ interface [56,57]. The refinement of atomic positions was unrestrained except for hydrogen atoms, which were maintained at riding positions. Appendix A (SM) summarizes the crystallographic data.

1,12-Dicarba-*closo*-dodecaborane-1-carboxylic acid (**1**) and 1,12-dicarba-*closo*-dodecaborane-1-carboxylic acid chloride (**2**) were prepared according to the literature [20].

Active esters: 3-(1,2-dicarba-*closo*-dodecaboran-1-yl)propionic acid *N*-succinimidyl ester (**5**), 3-(1,7-dicarba-*closo*-dodecaboran-1-yl)propionic acid *N*-succinimidyl ester (**6**), and 3-(1,12-dicarba-*closo*-dodecaboran-1-yl)propionic acid *N*-succinimidyl ester (**7**) were prepared according to the literature [21].

Aldehydes: 1-Formyl-1,2-dicarba-*closo*-dodecaborane (**11**), 1-formyl-1,7-dicarba-*closo*-dodecaborane (**12**), 1-formyl-1,12-dicarba-*closo*-dodecaborane (**13**), 2-(1,2-dicarba-*closo*-dodecaboran-1-yl)ethanal (**17**), 2-(1,7-dicarba-*closo*-dodecaboran-1-yl)ethanal (**18**), and 2-(1,12-dicarba-*closo*-dodecaboran-1-yl)ethanal (**19**) were prepared according to the literature [22].

#### 3.1.1. Synthesis of 1,12-dicarba-*closo*-dodecaborane-1-carboxylic acid hydrazide (3) 

1,12-Dicarba-*closo*-dodecaborane-1-carboxylic acid (**1**) (40 mg, 0.21 mmol) was dissolved in dry toluene (4 mL), and phosphorous pentachloride (47 mg, 0.21 mmol) was added. The reaction mixture was refluxed for 4 h under an inert atmosphere (Ar). Subsequently, the solvent was evaporated to dryness under vacuum. To the solution of 1,12-dicarba-*closo*-dodecaborane-1-carboxylic acid chloride (**2**) (44 mg, 0.21 mmol) in dry MeOH (10.5 mL), hydrazine hydrate (41 µL, 0.84 mmol) was added. The reaction mixture was refluxed for 5 h. Subsequently, the solvent was evaporated to dryness under vacuum, and the crude product was purified by column chromatography on silica gel (230–400 mesh) using a gradient elution from 0% to 9% MeOH in CH_2_Cl_2_. Next, product **3** was triturated with hexane (3 × 2 mL) in an ultrasonic bath to afford the product as a white solid. 

Yield: 41%, TLC (CH_2_Cl_2_/MeOH, 4:1, *v*/*v*): *R*_f_ = 0.58; ^1^H NMR (pyridine-d_5_, 600.17 MHz): *δ* (ppm) = 5.65 (br s, 2H, NH_2_), 3.27 (br s, 1H, CH_carborane_), 3.2–1.8 (m, 10H, B_10_H_10_); ^13^C NMR (pyridine-d_5_, 150.95 MHz): *δ* (ppm) = 89.61 (1C, C_carborane_), 62.22 (1C, CH_carborane_); ^11^B NMR (pyridine-d5, 192.59 MHz): *δ* (ppm) = 13.85 (s), 11.27 (s). FT-IR: *ν* (cm^−1^) = 3378 (NH), 3056 (CH_carborane_), 2602 (BH), 1591 (C=O), 719 (BB); APCI-MS: *m*/*z* 203 (100%), 171 (60%), calcd for C_3_H_13_B_10_ON_2_ = 202.21. 

#### 3.1.2. General Procedure for the Synthesis of Isonicotinyl Hydrazide **8**–**10**

Isoniazid (**4**) (16.6–21.9 mg, 0.12–0.16 mmol) was dissolved in absolute EtOH (1 mL) and cooled to 0 °C. Active esters **5**–**7** (1 eq.) were added. The reaction mixture was stirred for 4 days at 40 °C (for **8**) and for 3 days at RT (for **9** and **10**). Subsequently, the solvent was evaporated to dryness under vacuum, and products **8**–**10** were purified by column chromatography on silica gel (230–400 mesh) using a gradient of MeOH in CH_2_Cl (0–12%). Next, products **8**–**10** were triturated with hexane (3 × 2 mL) in an ultrasonic bath to afford a pure product. 

*N*′-((1,2-dicarba-*closo*-dodecaboran-1-yl)propanoyl)isonicotinohydrazide (**8**): white solid, yield: 56%. TLC (CH_2_Cl_2_/MeOH, 9:1, *v*/*v*): *R*_f_ = 0.48; ^1^H NMR (CD_3_OD, 600.26 MHz): *δ* (ppm) = 8.68 (dd, 2H, 3b, 3b′, *J*_HH_ = 6 Hz), 7.77 (dd, 2H, 2a, 2a′, *J*_HH_ = 6), 4.54 (br s, 1H, CH_carborane_), 2.65 (t, 2H, CH_2-β_)_,_ 2.54 (q, 2H, CH_2-α_), 2.5–1.7 (m, 10H, B_10_H_10_); ^13^C NMR (CD_3_OD, 150.95 MHz): *δ* (ppm) = 172.15 (1C, CO), 166.87 (1C, CO), 151.10 (2C, 3b, 3b′), 141.77 (1C, C1), 123.06 (2C, 2a, 2a′), 76.18 (1C, C_carborane_), 63.83 (1C, CH_carborane_), 33.84 (1C, CH_2_-linker), 33.70 (1C, CH_2_-linker); ^11^B{H BB} NMR (CD_3_OD, 192.59 MHz): *δ* (ppm) = −2.55 (s, 1B, B9), −5.85 (s, 1B, B12), −9.56 (s, 2B, B8, 10), −11.57 (s, 6B, B3, 4, 5, 6, 7, 11), −12.89 (s, 2B, B7, 11); FT-IR: *ν* (cm^−1^) = 3206 (NH), 3057 (CH_carborane_), 2603 (BH), 1698 (C=O), 1636 (C=O_amide_). 723 (BB); APCI-MS: *m*/*z* 336 (100%), calcd for C_11_H_21_B_10_N_3_O_2_ = 335,26; RP-HPLC (gradient A): *t*_R_ = 25.68 min.

*N*′-((1,7-dicarba-*closo*-dodecaboran-1-yl)propanoyl)isonicotinohydrazide (**9**): white solid, yield: 72%. TLC (CH_2_Cl_2_/MeOH, 9:1, *v*/*v*): *R*_f_ = 0.5; ^1^H NMR (CD_3_OD, 600.26 MHz): *δ* (ppm) = 8.71 (dd, 2H, 3b, 3b′, *J*_HH_ = 6 Hz), 7.80 (q, 2H, 2a, 2a′), 3.52 (br s, 1H, CH_carborane_), 2.9–1.7 (m, 10H, B_10_H_10_), 2.45–2.37 (m, 4H, CH_2-α_, CH_2-β_); ^13^C NMR (CD_3_OD, 150.95 MHz): *δ* (ppm) = 172.57 (1C, CO), 166.86 (1C, CO), 151.12 (2C, 3b, 3b′), 141.83 (1C, C1), 123.09 (2C, 2a, 2a′), 76.55 (1C, C_carborane_), 57.14 (1C, CH_carborane_), 34.65 (1C, CH_2_-linker), 33.01 (1C, CH_2_-linker); ^11^B{H BB} (CD_3_OD, 192.59 MHz): *δ* (ppm) = −4.34 (1B, B5), −9.75 (1B, B12), −10.87 (4B, B4, 6, 9, 10), −13.42 (2B, B8, 11), −15.05 (2B, B2, 3); FT-IR: *ν* (cm^−1^) = 3206 (NH), 3029 (CH_carborane_), 2595 (BH), 1698 (C=O), 1647 (C=O_amide_), 728 (BB); APCI-MS: *m*/*z* 336 (100%), calcd for C_11_H_21_B_10_N_3_O_2_ = 335.26; RP-HPLC (gradient A): *t*_R_ = 37.77 min. 

*N′*-((1,12-dicarba-*closo*-dodecaboran-1-yl)propanoyl)isonicotinohydrazide (**10**): white solid, yield: 67%. TLC (CH_2_Cl_2_/MeOH, 9:1, *v*/*v*): *R*_f_ = 0.86; ^1^H NMR (CD_3_OD, 600.26 MHz): *δ* (ppm) = 8.71 (dd, 2H, 3b, 3b′, *J*_HH_ = 6 Hz), 7.79 (q, 2H, 2a, 2a′), 3.16 (br s, 1H, CH_carborane_), 2.7–1.7 (m, 10H, B_10_H_10_), 2.23 (q, 2H, CH_2- α_)_,_ 2.05 (t, 2H, CH_2-ß_); ^13^C NMR (CD_3_OD, 150.95 MHz): *δ* (ppm) = 172.76 (1C, CO), 166.90 (1C, CO), 151.20 (2C, 3b, 3b′), 141.91 (1C, C1), 123.18 (2C, 2a, 2a′), 84.51 (1C, C_carborane_), 60.23 (1C, CH_carborane_), 35.05 (1C, CH_2_-linker), 34.29 (1C, CH_2_-linker); ^11^B{H BB} (CD_3_OD, 192.59 MHz): *δ* (ppm) = −19.63 (s), −20.37 (s); FT-IR: *ν* (cm^−1^) = 3208 (NH), 3011 (CH_carborane_), 2604 (BH), 1704 (C=O), 1646 (C=O_amide_), 730 (BB); APCI-MS: *m*/*z* 336 (100%), calcd for C_11_H_21_B_10_N_3_O_2_ = 335,26; RP-HPLC (gradient A): *t*_R_ = 30.70 min.

#### 3.1.3. General Procedure for the Synthesis of Isonicotinyl Hydrazide 14–16

Isoniazid (**4**) (9.1–20 mg, 0.07–0.145 mmol) was added in three portions over 3 h to a solution of aldehyde **11**–**13** (1 eq.) dissolved in dry ethyl acetate (**11**, 0.25 mL) or absolute EtOH (**12** and **13**, 0.25–0.3 mL). The reaction mixture was stirred for 20 h (for **14**), 96 h (for **15**), and 24 h (for **16**) at 40 °C. Subsequently, the solvent was evaporated to dryness under vacuum, and the crude products **14**–**16** were purified by silica gel (230–400 mesh) column chromatography using a gradient of MeOH in CH_2_Cl_2_ (0–10%) to afford pure products. Additionally, compound **16** was dissolved in a mixture of MeOH:H_2_O (30:1, *v*/*v*, 6.2 mL), and a Dowex 50WX8 Na^+^ form (obtained from H^+^ form, 150 mg) was added. The reaction mixture was stirred at RT overnight, and an additional portion of the same Dowex was added. The solution was stirred for the next 24 h. The solvents were evaporated to dryness under vacuum. The residue was suspended in MeOH in CH_2_Cl_2_ (10%), poured onto silica gel (230–400 mesh), and eluted from silica gel using a gradient of MeOH in CH_2_Cl_2_ (10–20%) to afford product **16**.

*N′*-((1,7-dicarba-*closo*-dodecaboran-1-yl)methylidene)isonicotinohydrazide (**14**): white solid, yield 84%. TLC (CH_2_Cl_2_/MeOH, 9:1, *v*/*v*): *R*_f_ = 0.3; ^1^H NMR (CD_3_OD, 600.26 MHz): *δ* (ppm) = 8.73 (dd, 2H, 3b, 3b′, *J*_HH_ = 5.6 Hz), 7.82 (dd, 2H, 2a, 2a′, *J*_HH_=5.6 Hz), 7.68 (s, 1H, N=CH), 3.70 (br s, 1H, CH_carborane_), 2.7–1.7 (m, 10H, B_10_H_10_); ^13^C NMR (CD_3_OD, 150.95 MHz): *δ* (ppm) = 164.35 (1C, CO), 151.14 (2C, 3b, 3b′), 146.49 (1C, N=CH), 142.02 (1C, C1), 123.02 (2C, 2a, 2a′), 74.30 (1C, C_carborane_), 57.50 (1C, CH_carborane_); ^11^B{H BB} NMR (CD_3_OD, 192.59 MHz): *δ* (ppm) = −5.09 (s, 1B, B5), −7.76 (s, 1B, B12), −10.69 (s, 4B, B4, 6, 9, 10), −13.06 (s, 2B, B8, 11), −15.18 (s, 2B, B2, 3); FT-IR: ν (cm^−1^) = 3197 (NH), 3018 (CH_carborane_), 2605 (BH), 1660 (C=O), 1552 (C=N), 724 (BB); APCI-MS: *m*/*z* 292 (100%), calcd for C_9_H_17_B_10_N_3_O = 292.23; RP-HPLC (gradient A): *t*_R_ = 24.85 min.

*N′*-((1,12-dicarba-*closo*-dodecaboran-1-yl)methylidene)isonicotinohydrazide (**15**): white solid, yield 57%. TLC (CH_2_Cl_2_/MeOH, 9:1, *v*/*v*): *R*_f_ = 0.35; ^1^H NMR (CD_3_OD, 600.26 MHz): *δ* (ppm) = 8.71 (dd, 2H, 3b, 3b′, *J*_HH_ = 4.5 Hz), 7.77 (dd, 2H, 2a, 2a′, *J*_HH_ = 5.8 Hz), 7.40 (s, 1H, N=CH), 3.35 (br s, 1H, CH_carborane_), 2.7–1.7 (m, 10H, B_10_H_10_); ^13^C NMR (CD_3_OD, 150.95 MHz): *δ* (ppm) = 164.32 (1C, CO), 151.14 (2C, 3b, 3b′), 147.55 (1C, N=CH), 142.15 (1C, C1), 123.05 (2C, 2a, 2a′), 81.26 (1C, C_carborane_), 62.88 (1C, CH_carborane_); ^11^B{H BB} NMR (CD_3_OD, 192.59 MHz): *δ* (ppm) = −13.09 (s, 5B), −14.83 (s, 5B); FT-IR: ν (cm^−1^) = 3189 (NH), 2924 (CH_carborane_), 2610 (BH), 1656 (C=O), 1551 (C=N), 728 (BB); APCI-MS: *m*/*z* 292 (100%), calcd for C_9_H_17_B_10_N_3_O = 292.23; RP-HPLC (gradient A): *t*_R_ = 23.03 min. 

*N′*-((7,8-dicarba-*nido*-undecaboranyl)methylidene)isonicotinohydrazide (**16**): pale yellow solid, yield 50%. TLC (CH_2_Cl_2_/MeOH, 9:1, *v*/*v*): *R*_f_ = 0.1; ^1^H NMR (CD_3_OD, 600.26 MHz): *δ* (ppm) = 8.67 (dd, 2H, 3b, 3b′, *J*_HH_ = 5.9 Hz), 7.80 (dd, 2H, 2a, 2a′, *J*_HH_ = 4.6 Hz), 7.61 (s, 1H, N=CH), 2.65 (br s, 1H, CH_carborane_), 2.25–1.75 (m, 10H, B_9_H_11_), −2.67 (br s, 1H, the bridging H atom); ^13^C NMR (CD_3_OD, 150.95 MHz): *δ* (ppm) = 164.06 (1C, CO), 161.62 (1C, N=CH), 151.26 (2C, 3b, 3b′), 143.24 (1C, C1), 123.45 (2C, 2a, 2a′); ^11^B{H BB} NMR (CD_3_OD, 192.59 MHz): *δ* (ppm) = −9,50 (s, 2B, B5, 9), −12.56 (s, 1B, B2), −15.32 (s, 1B, B11), −16.96 (s, 1B, B3), −21.26 (s, 1B, B4), −21.95 (s, 1B, B6), −32.91 (s, 1, B10), −35.70 (s, 1B, B1); FT-IR: ν (cm^−1^) = 3307 (NH), 2922 (CH_carborane_), 2515 (BH), 1651 (C=O), 1541 (C=N) 685 (BB); APCI-MS: *m*/*z* 280 (100%), calcd for C_9_H_17_B_9_N_3_O = 281.22; RP-HPLC (gradient A): *t*_R_ = 33.14 min.

#### 3.1.4. General Procedure for the Synthesis of Isonicotinyl Hydrazide 20–22

Isoniazid (**4**) (13.3–36.8 mg, 0.097–0.27 mmol) was added in three portions over 3 h to a solution of aldehyde **17**–**19** (1 eq.) dissolved in absolute EtOH (0.4–0.7 mL). The reaction mixture was stirred for 12 h (for **20**) at 35 °C, 12 h (for **21**) at RT, and 24 h (for **22**) at 40 °C. Subsequently, the solvent was evaporated to dryness under vacuum, and the crude products **20**–**22** were purified by silica gel (230–400 mesh) column chromatography using a gradient of MeOH in CH_2_Cl_2_ (0–10%) to afford pure products. 

*N′*-((1,2-dicarba-*closo*-dodecaboran-1-yl)ethylidene)isonicotinohydrazide (**20**): pale orange solid, yield 67%. TLC (CH_2_Cl_2_/MeOH, 9:1, *v*/*v*): *R*_f_ = 0.27; ^1^H NMR (CD_3_OD, 600.26 MHz): *δ* (ppm) = 8.73 (dd, 2H, 3b, 3b′, *J*_HH_ = 4.5, 1.7 Hz), 7.84 (dd, 2H, 2a, 2a′, *J*_HH_ = 4.5, 1.5 Hz), 7.70 (t, 1H, N=CH, *J*_HH_ = 5.9 Hz), 4.69 (br s, 1H, CH_carborane_), 3.34 (d, 2H, CH_2_-linker, *J*_HH_ = 6.0 Hz), 2.7–1.7 (m, 10H, B_10_H_10_); ^13^C NMR (CD_3_OD, 150.95 MHz): *δ* (ppm) = 164.77 (1C, CO), 151.29 (2C, 3b, 3b′), 149.40 (1C, N=CH), 142.22 (1C, C1), 123.25 (2C 2a, 2a′), 73.33(1C, C_carborane_), 63.54 (1C, CH_carborane_), 40.94 (1C, CH_2_-linker); ^11^B{H BB} NMR (CD_3_OD, 192.59 MHz): *δ* (ppm) = −2.34 (s, 1B, B9), −5.37 (s, 1B, B12), −9.24 (s, 2B, B8, 10), −11.60 (s, 2B, B3, 4), −12.06 (s, 2B. B5, 6), −12.74 (s, 2B, B7, 11); FT-IR: ν (cm^−1^) = 3207 (NH), 3048 (CH_carborane_), 2586 (BH), 1663 (C=O_amide_), 1552 (C=N), 722 (BB); APCI-MS: *m*/*z* 306 (100%), calcd for C_10_H_19_B_10_N_3_O = 306.25; RP-HPLC (gradient B): *t*_R_ = 21.82 min.

*N′*-((1,7-dicarba-*closo*-dodecaboran-1-yl)ethylidene)isonicotinohydrazide (**21**): white solid, yield 79%. TLC (CH_2_Cl_2_/MeOH, 9:1, *v*/*v*): *R*_f_ = 0.3; ^1^H NMR (CD_3_OD, 600.26 MHz): *δ* (ppm) = 8.73 (dd, 2H, 3b, 3b′, *J*_HH_ = 4.5, 1.7 Hz), 7.83 (dd, 2H, 2a, 2a′, *J*_HH_ = 4.5, 1.7 Hz), 7.62 (t, 1H, N=CH, *J*_HH_ = 6.0 Hz), 3.60 (br s, 1H, CH_carborane_), 3.04 (d, 2H, CH_2_-linker, *J*_HH_ = 6.0 Hz), 2.7–1.7 (m, 10H, B_10_H_10_); ^13^C NMR (CD_3_OD, 150.95 MHz): *δ* (ppm) = 164.53 (1C, C=O), 151.13 (2C, 3b, 3b′), 150.71 (1C, N=CH), 142.18 (1C, C1), 123.10 (2C, 2a, 2a′), 57.61 (1C, CH_carborane_), 40.03 (1C, CH_2_-linker); ^11^B{H BB} NMR (CD_3_OD, 192.59 MHz): *δ* (ppm) = −4.45 (s, 1B, B5), −9.33 (s, 1B, B12), −10.58 (s, 2B, B4, 6), −10.99 (s, 2B, B9, 10), −13.19 (s, 2B, B8, 11), −14.98 (s, 2B, B2, 3); FT-IR: ν (cm^−1^) = 3200 (NH), 3046 (CH_carborane_), 2592 (BH), 1656 (C=O_amide_), 1550 (C=N), 728 (BB); APCI-MS: *m*/*z* 306 (100%), calcd for C_10_H_19_B_10_N_3_O =306.25; RP-HPLC (gradient B): *t*_R_ = 22.62 min.

*N′*-((1,12-dicarba-*closo*-dodecaboran-1-yl)ethylidene)isonicotinohydrazide (**22**): white solid, yield 55%. TLC (CH_2_Cl_2_/MeOH, 9:1, *v*/*v*): *R*_f_ = 0.4; ^1^H NMR (CD_3_OD, 600.26 MHz): *δ* (ppm) = 8.72 (dd, 2H, 3b, 3b′, *J*_HH_ = 6.0 Hz), 7.82 (dd, 2H, 2a, 2a′, *J*_HH_ = 4.6, 1.5 Hz), 7.44 (t, 1H, N=CH, *J*_HH_ = 6.0 Hz), 3.22 (br s, 1H, CH_carborane_), 2.71 (d, 2H, CH_2_-linker, *J*_HH_ = 6.0 Hz), 2.5–1.7 (m, 10H, B_10_H_10_); ^13^C NMR (CD_3_OD, 150.95 MHz): *δ* (ppm) = 164.62 (1C, CO), 151.25 (2C, 3b, 3b′), 150.87 (1C, N=CH), 142.35 (1C, C1), 123.25 (2C, 2a, 2a′), 61.05 (1C, CH_carborane_), 41.98 (1C, CH_2_-linker); ^11^B{H BB} NMR (CD_3_OD, 192.59 MHz): *δ* (ppm) = −12.67 (s, 5B), −14.70 (s, 5B); FT-IR: ν (cm^−1^) = 3208 (NH), 3039 (CH_carborane_), 2604 (BH), 1658 (C=O_amide_), 1553 (C=N), 732 (BB); APCI-MS: *m*/*z* 306 (100%), calcd for C_10_H_19_B_10_N_3_O =306.25; RP-HPLC (gradient B): *t*_R_ = 21.26 min.

#### 3.1.5. General Procedure for the Synthesis of Isonicotinyl Hydrazides **23**–**25**


A solution of methanolic HCl (5 M, 80–140 µL) was added dropwise to a solution of compounds **20**–**22** (10–13.08 mg, 0.033–0.043 mmol) and NaBH_3_CN (0.68 eq.) in MeOH (0.3–1 mL) until pH 3–5 was reached. The reaction mixture was stirred for 4 h at RT, and a solution of sodium bicarbonate was added at pH 7. MeOH was evaporated under reduced pressure. The residue was extracted with ethyl acetate (3 × 5 mL). The organic phase was separated, dried over MgSO_4_, filtered, and evaporated to dryness. The crude products **23**–**25** were purified by column chromatography on silica gel (230–400 mesh) using a gradient of MeOH in CHCl_3_ (0 to 10%) to afford a pure product. 

*N*′-((1,2-dicarba-*closo*-dodecaboran-1-yl)ethyl)isonicotinohydrazide (**23**): white solid, yield 53%. TLC (CH_2_Cl_2_/MeOH, 9:1, *v*/*v*): *R*_f_ = 0.27; ^1^H NMR (CD_3_OD, 600.26 MHz): *δ* (ppm) = 8.69 (dd, 2H, 3b, 3b′, *J*_HH_ = 5.7 Hz), 7.74 (dd, 2H, 2a, 2a′, *J*_HH_ = 4.5 Hz), 3.06 (t, 2H, CH_2_-linker, *J*_HH_ = 7.4 Hz), 2.52 (t, 2H, CH_2_-linker, *J*_HH_ = 7.4 Hz), 2.5–1.5 (m, 10H, B_10_H_10_), CH_carborane_ signal overlapped with signal from H_2_O in CD_3_OD; ^13^C NMR (CD_3_OD, 150.95 MHz): *δ* (ppm) = 166.93 (1C, CO), 150.94 (2C, 3b, 3b′), 142.38 (1C, C1), 122.75 (2C, 2a, 2a′), 75.14 (1C, C_carborane_), 63.28 (1C, CH_carborane_), 51.11 (1C, CH_2_-linker), 36.41 (1C, CH_2_-linker); ^11^B{H BB} NMR (CD_3_OD, 192.59 MHz): *δ* (ppm) = −2.71 (s, 1B, B9), −5.76 (s, 1B, B12), −9.71 (s, 2B, B8, 10), −11.24 (s, 2B, B3, 4), −11.96 (s, 2B, B5, 6), −12.96 (s, 2B, B7, 11); FT-IR: ν(cm^−1^) = 3248 (NH), 2578 (BH), 1660 (C=O), 722 (BB). APCI-MS: *m*/*z* 308 (100%), calcd for C_10_H_21_B_10_N_3_O = 307.27; RP-HPLC (gradient B): *t*_R_ = 21.73 min.

*N*′-((1,7-dicarba-*closo*-dodecaboran-1-yl)ethyl)isonicotinohydrazide (**24**): white solid, yield 80%. TLC (CH_2_Cl_2_/MeOH, 9:1, *v*/*v*): *R*_f_ = 0.3; ^1^H NMR (CD_3_OD, 600.26 MHz): *δ* (ppm) = 8.69 (dd, 2H, 3b, 3b′, *J*_HH_ = 4.6 Hz), 7.73 (dd, 2H, 2a, 2a′, *J*_HH_ = 4.5 Hz), 3.51 (br s, 1H, CH_carborane_), 2.95–2.92 (m, 2H, CH_2_-linker), 2.75–1.75 (m, 10H, B_10_H_10_), 2.26–2.23 (m, 2H, CH_2_-linker); ^13^C NMR (CD_3_OD, 150.95 MHz): *δ* = 166.05 (1C, CO), 151.37 (2C, 3b, 3b′), 142.78 (1C, C1), 123.13 (2C, 2a, 2a′), 75.53 (1C, C_carborane_), 57.41 (1C, CH_carborane_), 52.47 (1C, CH_2_-linker), 36.24 (1C, CH_2_-linker); ^11^B{H BB} NMR (CD_3_OD, 192.59 MHz): *δ* (ppm) = −4.28 (s, 1B, B5), −9.72 (s, 1B, B12), −10.90 (s, 4B, B4, 6, 9, 10), −13.49 (s, 2B, B8, 11), −15.08 (s, 2B, B2, 3); FT-IR: ν(cm^−1^) = 3254 (NH), 3044 (CH_carborane_), 2592 (BH), 1647 (C=O), 729 (BB); APCI-MS: *m*/*z* 308 (100%), calcd for C_10_H_21_B_10_N_3_O = 307.27; RP-HPLC (gradient B): *t*_R_ = 22.18 min.

*N*′-((1,12-dicarba-*closo*-dodecaboran-1-yl)ethyl)isonicotinohydrazide (**25**): white solid, yield 65%. TLC (CH_2_Cl_2_/MeOH, 9:1, *v*/*v*): *R*_f_ = 0.4; ^1^H NMR (CD_3_OD, 600.26 MHz): *δ* = 8.68 (dd, 2H, 3b, 3b′, *J*_HH_ = 4.6 Hz), 7.71 (dd, 2H, 2a, 2a′, *J*_HH_ = 4.5 Hz), 3.15 (br s, 1H, CH_carborane_), 2.74–2.72 (m, 2H, CH_2_-linker), 2.5–1.75 (m, 10H, B_10_H_10_), 1.93–1.90 (m, 2H, CH_2_-linker); ^13^C NMR (CD_3_OD, 150.95 MHz): *δ* = 166.66 (1C, CO), 151.08 (2C, 3b, 3b′), 142.50 (1C, C1), 122.82 (2C, 2a, 2a′), 83.25 (1C, C_carborane_), 60.34 (1C, CH_carborane_), 51.74 (1C, CH_2_-linker), 37.80 (1C, CH_2_-linker); ^11^B{H BB} NMR (CD_3_OD, 192.59 MHz): *δ* = −19.62 (s, 5B), −20.38 (s, 5B); FT-IR: ν(cm^−1^) = 3255 (NH), 2925 (CH_carborane_), 2601 (BH), 1646 (C=O_amide_), 729 (BB). APCI-MS: *m*/*z* 308 (100%), calcd for C_10_H_21_B_10_N_3_O = 307.27; RP-HPLC (gradient B): *t*_R_ = 22.19 min.

### 3.2. Biology

#### 3.2.1. Bacterial Strain and Growth Conditions

The *Mtb* H37Rv and mutant Δ*katG* strains used in this study were cultured in Middlebrook 7H10 medium supplemented with 10% OADC (albumin–dextrose–sodium chloride) and 0.5% glycerol. The liquid cultures were grown in Middlebrook 7H9 broth with 10% OADC and 0.05% Tween-80 (pH 7.0) and were supplemented with chemicals at various concentrations when required.

#### 3.2.2. M. Tuberculosis Susceptibility Tests 

The *Mtb* H37Rv strain was grown in Middlebrook 7H9 broth supplemented with 10% OADC and 0.05% Tween-80 (pH 7) for 4–6 days until an OD_600_ of 1 was reached. Then, the bacterial culture was suspended in Middlebrook 7H9 at an OD_600_ of approximately 0.1. The cultures were supplemented with compounds and grown for 96 h at 37 °C. The growth was monitored by OD_600_ and colony-forming unit (CFU) analyses. At each 24-h interval, culture samples (100 µL) were withdrawn and used to perform serial dilutions. The dilutions were plated on Middlebrook 7H10 agar plates and incubated at 37 °C for 21 days. The colonies were then counted to determine the bacterial cell count (CFU/mL). 

The tested compounds were dissolved in DMSO and added directly to the growth medium. The final concentration of DMSO in the medium never exceeded 0.1% (*v*/*v*) and had no effect on the growth of *Mtb*. The experiment was performed in 3 repetitions. 

#### 3.2.3. Construction of Gene Replacement Vector and Disruption of the *Mtb katG* Gene at its Native Chromosomal Loci

Suicidal delivery vectors carrying nonfunctional *rv1908c* (*katG*) were prepared in three steps. First, the 5′ upstream region of *katG* (1194bp) was cloned into a suicidal recombination delivery vector, p2Nil [58]. Next, the 3′ fragment of *katG* and its downstream region (1619 bp) was ligated with plasmid from step 1 to create a truncated, out-of-frame copy of the gene. Finally, a 6-kb PacI cassette from pGOAL17 was added, resulting in the suicidal delivery vector pKatG-0, which was used to engineer the directed *Mtb* mutant strain.

A two-step protocol, based on a homologous recombination, was applied to generate a defined mutant strain lacking the functional KatG protein, as described previously [58,59]. The suicidal recombination plasmid pKatG-0 was integrated into the *Mtb* chromosome by homologous recombination. The obtained single-crossover (SCO) recombinants were blue, Kan^R^, and sensitive to sucrose. The SCO strains were further processed to select for double-crossover (DCO) mutants that were white, Kan^S^, and resistant to sucrose (2%) and 25 µg/mL of INH (**4**). The genotypes of the obtained mutant DCO strains were confirmed by Southern blot hybridization using the Amersham ECL Direct Nucleic Acid Labeling and Detection System (GE Healthcare, Chicago, IL, USA) following the manufacturer′s instructions. The hybridization probe was generated by PCR, as shown in Appendix A (SM). 

#### 3.2.4. Cell Culture

The cytotoxic properties of the tested compounds were evaluated using the human immortalized cell line HaCaT established from human keratinocytes. The cell line was purchased from CLS and was grown in Dulbecco′s modified Eagle′s medium (DMEM) (Sigma-Aldrich) and supplemented with 10% heat-inactivated fetal bovine serum (FBS; Sigma-Aldrich) and antibiotics (Sigma-Aldrich). The cells were incubated at 37 °C in a humidified atmosphere containing 5% CO_2_.

#### 3.2.5. MTT Analysis of Cell Viability

HaCaT cells were seeded into 96-well plates at a density of 12,500 cells per well and incubated overnight at 37 °C in a humidified atmosphere containing 5% CO_2_. To estimate the cellular toxicity, the culture medium was removed and replaced with a freshly prepared solution of the compounds in culture medium or the medium itself as the control group. Stock solution of each compound was prepared in DMSO at the final concentration of 20 mM. Cytotoxicity was evaluated by the MTT assay for previously determined MIC_50_ and MIC_99_ values for the tested compounds. This colorimetric assay used a reduction of the yellow tetrazolium salt (3-(4,5-dimethylthiazol-2-yl)-2,5-diphenyltetrazolium bromide or MTT; Sigma-Aldrich) to measure the cellular metabolic activity as a reflection of cell viability. The cells were incubated in the presence of 0.5-mg MTT/mL (final concentration) for 1 h at 37 °C. After incubation, formazan crystals were dissolved in DMSO (BioShop, Burlington, Canada). The absorbance was read at 570 nm. The amount of produced formazan was proportional to the number of live and metabolically active cells. The experimental points were represented as a mean from three replicate experiments with standard deviations. 

#### 3.2.6. Determination of the CC_50_ Value

To determine the CC_50_ value, HaCaT cells (1.25 × 10^4^) were seeded onto 96-well plates and incubated overnight at 37 °C in a humidified atmosphere containing 5% CO_2_. Subsequently, the growth medium was removed and replaced with a solution of the compounds in a concentration range up to 1 mM. Control cells were grown in culture medium alone. Cell viability was evaluated by the MTT assay, as described above. Each experiment consisted of 6 replications of each concentration and two separate repetitions. Cell viability was expressed as a percentage of the absorbance of control cells, which were considered as to have 100% absorbance. The 50% cytotoxic concentration (CC_50_) was defined as the concentration required to reduce the cell growth by 50% compared to untreated controls. The CC_50_ value was calculated using linear regression from the plotted cell survival data.

To calculate the selectivity index (SI), the CC_50_ value was divided by the MIC_99_ of *Mtb* or of Δ*katG* (SI ratio = CC_50_/MIC_99_) for each compound; CC_50_ values and MIC_99_ values were used from Table 2.

#### 3.2.7. Apoptosis/Necrosis Analysis by Flow Cytometry

The apoptosis/necrosis assay was performed by double-staining of cells with YO-PRO-1 (Thermo Fisher Scientific, Waltham, MA, USA) and propidium iodide (PI, Sigma-Aldrich) fluorescent dyes. Briefly, HaCaT cells (4.5 × 10^5^) were seeded onto 6-well plates. On the next day, the cells were treated for 24 h with the analyzed compounds at a concentration corresponding to the previously determined MIC_50_ and MIC_99_ values. Subsequently, the cells were detached with trypsin (Thermo Fisher Scientific), washed twice with DPBS (1 mL) (Thermo Fisher Scientific), and stained with YO-PRO-1 and PI according to the manufacturer′s protocol for 30 min at 37 °C in dark. The cells were analyzed immediately after staining with 488-nm excitation by FACSCalibur (Becton Dickinson, Franklin Lakes, NJ, USA), and the data were analyzed by FlowJo software.

### 3.3. Physicochemical Investigation

#### 3.3.1. Partition (P) and Distribution (D) Coefficient Measurements 

The assay was based on the shake-flask procedure [46]. To a solution (100 µM, 1 mL) of unmodified INH (**4**), modified derivatives (**8**–**10**, **14**–**15**, and **20**–**25**) in water-saturated octan-1-ol (for derivative **16** in PBS-saturated octan-1-ol), containing 0.65–2.2% of MeOH (HPLC purity) in a 2-mL Eppendorf tube, octan-1-ol-saturated water/PBS (1 mL) was added. The resulting mixture was shaken vigorously at RT for 0.5 h, and the mixture was then allowed to stand for 1 h for phase separation at RT. Each sample was subsequently centrifuged at 13,000 rpm for 10 min, and 0.15 mL of the water and organic solution was then transferred into a 96-well plate. The absorption was measured at λ = 265 (**4**), 265 (**8**), 265 (**9**), 265 (**10**), 263 (**14**), 263 (**15**), 265 (**16**), 261 (**20**), 262 (**21**), 261 (**22**), 263 (**23**), 265 (**24**), and 265 (**25**) nm. *P* or *D*_7.4_ measurements were performed in nine replicates. The following formula was used to determine the log *P* or log *D*_7.4_:(1)$$\log P\mathrm{or}\log D_{7.4}=\log_{10}\frac{{\left[c\right]}_{\mathrm{for}\;1-\mathrm{octanol}\;\mathrm{sample}\;}}{\;{\left[c\right]}_{\mathrm{for}\;\mathrm{water}/\mathrm{buffer}\;\mathrm{sample}}}$$

#### 3.3.2. Parallel Artificial Membrane Permeability Assay

Parallel artificial membrane permeability assay (PAMPA) was performed using MultiScreen Filter Plate (MAIPNTR10) and MultiScreen Acceptor Plate (MSSACCEPTOR) (Merck Millipore, Warsaw, Poland). The tested compounds and reference compound (propranolol) were dissolved in 0.01-M PBS (pH 7.4) buffer containing 5% MeOH or 5% DMSO (**4**, **8**–**10**), to the final concentration of 100 µM or 800 µM (**4**, **8**–**10**, and **23**). The acceptor 96-well microplate was filled with 0.01-M PBS (300 µL, pH 7.4) containing 5% MeOH or 5% DMSO, and the filter surface of the donor microplate was impregnated with lecithin (5 µL, 2% in dodecane). Then, solutions of the tested compounds and reference compound were added to the donor plate (150 µL). The donor filter plate was carefully placed on the acceptor plate. The lid was placed on the plate, and the entire plate sandwich was placed into a closed container with a wet towel along the bottom to circumvent evaporation during the incubation process. The container was placed on a shaker for agitation at approximately 100 rpm for 18 h at RT. After incubation, the donor plate was carefully removed, and the concentration of the tested compounds in both compartments was determined using a UV-Vis spectrophotometer. Each sample was analyzed at the wavelength mentioned in Section 3.3.1 (wavelength 289 nm for propranolol). *P*_e_ measurement was performed in five replicates. The following formula was used to determine the log *P*_e_: (2)$$\log\;P_{e}=\log\{C\;\times \;-\ln(1-\frac{{\left[\mathrm{Drug}\right]}_{A}}{{\left[\mathrm{Drug}\right]}_{E}})\};\mathrm{where}\;C=\left(\frac{V_{A}\times \;V_{D}\;\;\;\;\;\;\;\;\;\;\;}{\left(V_{D}+\;V_{A}\right)\mathrm{area}\;\times \mathrm{time}}\right)\;\;\;\;\;$$
where V_D_ is the volume of the donor well (150 μL), V_A_ is the volume in the acceptor well (300 μL), area is the active surface area of the membrane (0.283 cm^2^), time is the incubation time of the assay in seconds, (Drug)_A_ is the absorbance of the compound in the acceptor well after the incubation period, and (Drug)_E_ is the absorbance of the compound at the concentration of the theoretical equilibrium. The ability of the tested compounds to permeate the artificial membranes was classified according to a previous study [54]. Impermeable compounds have a log *P*_e_ of > –6.14, low-permeability compounds have a log *P*_e_ between > –6.14 and < –5.66, medium-permeability compounds have a log *P*_e_ between > –5.66 and < –5.33, and high-permeability compounds have a log *P*_e_ of > –5.33. 

## 4. Conclusions

In the search for new antitubercular agents, we developed a method for the synthesis of INH modified with a *closo*-*ortho*-/*meta*-/*para*-carborane and *nido*-carborane with good yields. The present approach can provide a new route for generating for the first time INH derivatives with various boron clusters. 

Selected compounds, containing *closo*-carborane, exhibited significant activity, and one modified with *nido*-carborane exhibited potent activity, similar to INH, against *Mtb* in vitro. The presence of a carborane cluster, regardless of its structure, contributed significantly to increasing INH–carborane hybrids activity against the Δ*katG* mutant, in comparison to INH. Additionally, low cytotoxicity of the chosen hybrids combined with their activity against the *Mtb* or the Δ*katG* mutant make them a hit compounds that can be developed into a lead candidate for further studies.

The present work demonstrated that the conjugation of the biological active of isoniazid and the inorganic boron cluster, with their properties, are useful in drug design and might enable the development of a novel class of hybrids, lead compounds, with potential antimycobacterial activity. Studies on the development of new hybrids of isoniazid containing boron clusters are currently in progress in our laboratory.

## Figures and Tables

**Figure 1 pharmaceuticals-13-00465-f001:**
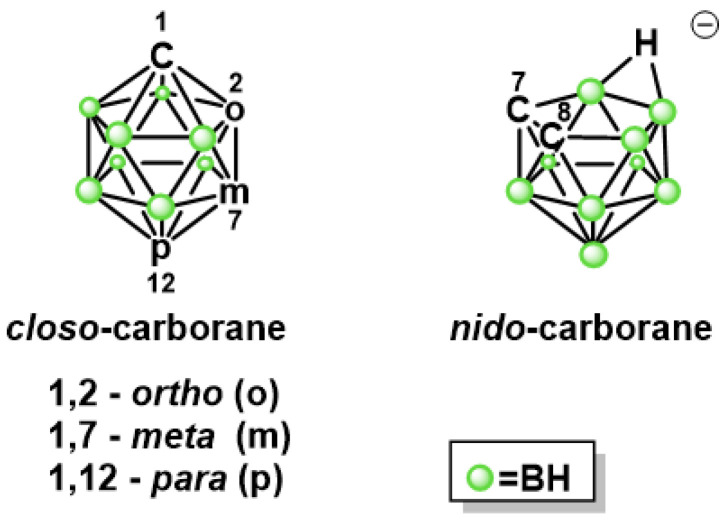
General structure of icosahedral dicarba-*closo*-dodecaborane (*closo*-carborane, C_2_B_10_H_12_) and 7,8-dicarba-*nido*-undecaborate anion (*nido*-carborane, 7,8-C_2_B_9_H_11_(–1)).

**Figure 2 pharmaceuticals-13-00465-f002:**
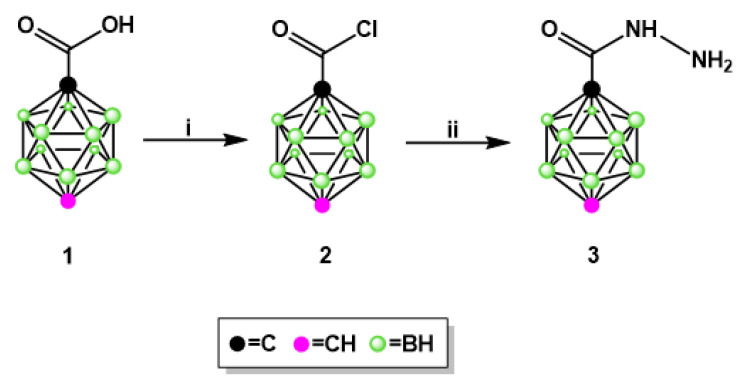
Synthesis of compound 3—analog of INH: (**i**) PCl5, reflux 4 h and (**ii**) NH_2_-NH_2_, reflux 5 h.

**Figure 3 pharmaceuticals-13-00465-f003:**
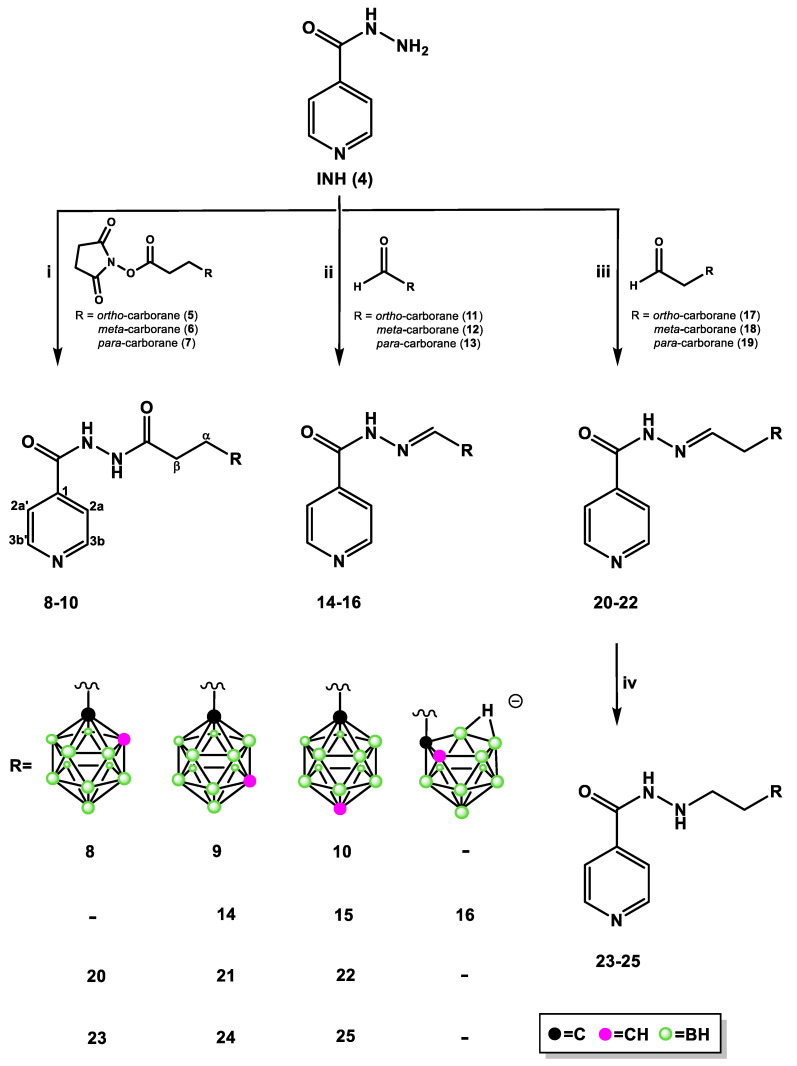
INH-carborane hybrids: (**i**) 8: 3-(1,2-dicarba-*closo*-dodecaboran-1-yl)propionic acid *N*-succinimidyl ester (**5**), EtOH_abs_, 4 days, 40 °C; 9: 3-(1,7-dicarba-*closo*-dodecaboran-1-yl)propionic acid *N*-succinimidyl ester (**6**), EtOH_abs_, 3 days, room temperature (RT); 10: 3-(1,12-dicarba-*closo*-dodecaboran-1-yl)propionic acid *N*-succinimidyl ester (**7**), EtOH_abs_, 3 days, RT; (**ii**) 14: 1-formyl-1,7-dicarba-*closo*-dodecaborane (**12**), EtOH_abs_, 20 h, 40 °C; 15: 1-formyl-1,12-dicarba-*closo*-dodecaborane (13), EtOH_abs_, 96 h, 40 °C; 16: 1-formyl-1,2-dicarba-*closo*-dodecaborane (**11**), ethyl acetate_anh_, 24 h, 40 °C; (**iii**) 20: 2-(1,2-dicarba-*closo*-dodecaboran-1-yl)ethanal (17), EtOH_abs_, 12 h, 35 °C; 21: 2-(1,7-dicarba-*closo*-dodecaboran-1-yl)ethanal (**18**), EtOH_abs_, 12 h, RT; 22: 2-(1,12-dicarba-*closo*-dodecaboran-1-yl)ethanal (**19**), EtOH_abs_, 24 h, 40 °C; (**iv**) 23–25: *N*′-((1,2-dicarba-*closo*-dodecaboran-1-yl)ethylidene)isonicotinohydrazide (**20**)/*N*′-((1,7-dicarba-*closo*-dodecaboran-1-yl)ethylidene)isonicotinohydrazide (**21**)/*N*′-((1,12-dicarba-*closo*-dodecaboran-1-yl)ethylidene)isonicotinohydrazide (**22**), NaBH_3_CN, MeOH_anh_, HCl (5 M), 4 h, RT.

**Figure 4 pharmaceuticals-13-00465-f004:**
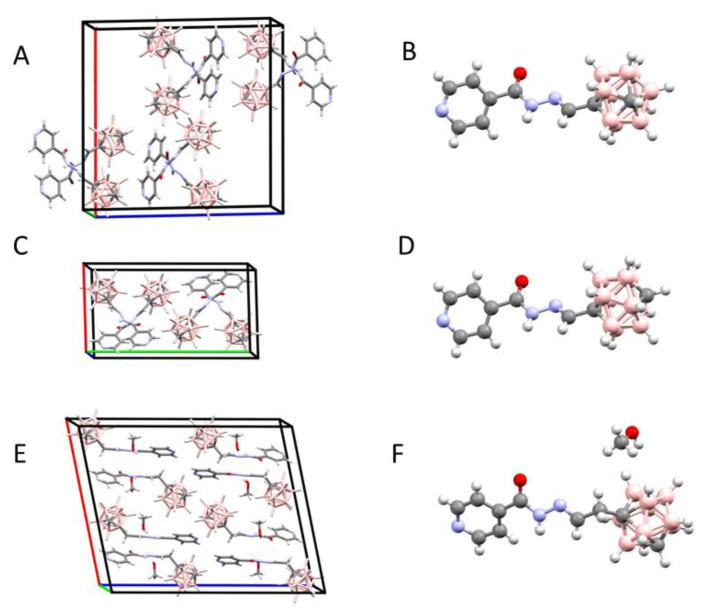
The crystallographic structures of INH–carborane hybrids: crystal packing and molecular structure of **14** (**A**,**B**), **15** (**C**,**D**), and **21** (**E**,**F**).

**Figure 5 pharmaceuticals-13-00465-f005:**
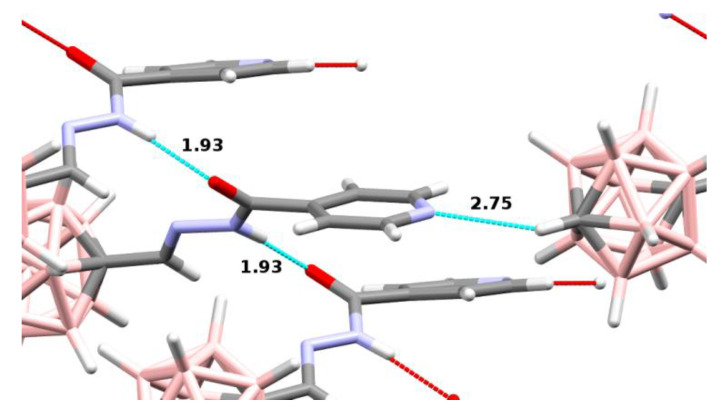
Intermolecular interactions in the crystal structure of **15**. Hydrogen bonds are indicated by dotted lines, and distances are in Å. Some molecules are hidden for clarity.

**Figure 6 pharmaceuticals-13-00465-f006:**
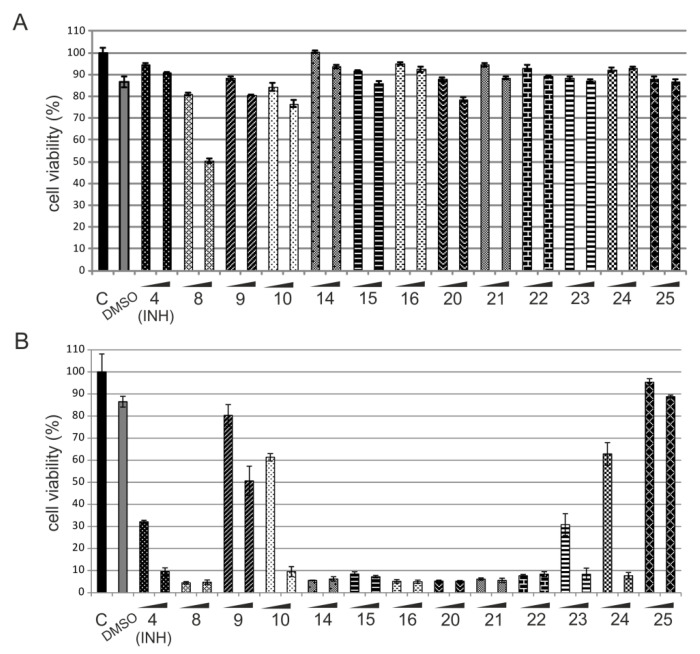
Cell viability analysis of HaCaT cells treated with the analyzed compounds. HaCaT cells were treated for 24 h with the compounds at concentrations that correspond to the minimum inhibitory concentration (MIC)_50_ and MIC_99_ values obtained for both strains: wild-type *Mtb* (**A**) and Δ*katG* (**B**). Cell viability was analyzed by the (3-(4,5-dimethylthiazol-2-yl)-2,5-diphenyltetrazolium bromide (MTT) tetrazolium salt assay.

**Figure 7 pharmaceuticals-13-00465-f007:**
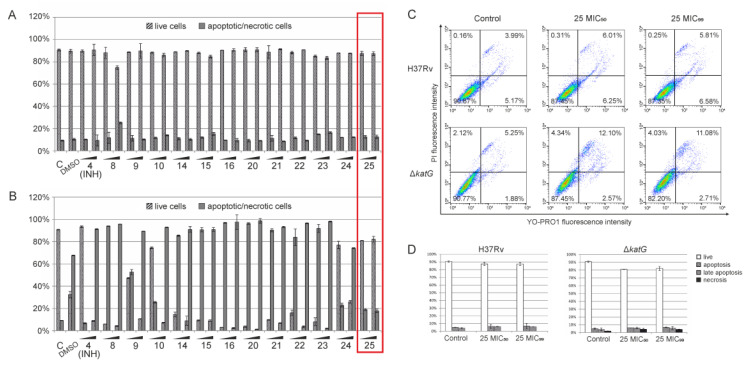
Flow cytometry analysis of apoptosis/necrosis in HaCaT cells after treatment with compounds. Cell toxicity was analyzed using the apoptosis/necrosis assay by flow cytometry. The cells were simultaneously incubated with the compounds for 24 h at the desired concentrations that were estimated previously for both wild-type (**A**) and mutant strains (**B**), corresponding to the MIC_50_ and MIC_99_ values. (**C**) Simultaneous flow cytometry analysis of apoptosis/necrosis in cells treated with compound **25** at concentrations determined as MIC_50_ and MIC_99_ for the wild-type and mutant strains. (**D**) Flow cytometry data for compound **25** were collected and plotted on bar graphs. Data are presented as mean ± SD.

**Table 1 pharmaceuticals-13-00465-t001:** Weak H-bonding interactions in the crystal lattice between the carborane C-H groups and the pyridine N atoms of the neighboring molecules.

Compound	C-N/H-N Distances (Å)	C-H-N Angle (°)
**14**	3.33/2.39	142
**15**	3.49/2.75	124
**21**	3.50/2.65	134

**Table 2 pharmaceuticals-13-00465-t002:** In vitro antimycobacterial activity of compounds **8**–**10**, **14**–**16**, and **20**–**25** against *Mtb* and Δ*katG*; their cytotoxicity; and their selectivity indexes.

Compound	*Mtb*		*katG*		HaCaT	SI	
	MIC_50_ ^a^ (µM)	MIC_99_ ^b^ (µM)	MIC_50_ (µM)	MIC_99_ (µM)	CC_50_ ^c^ ± SD (µM)	*Mtb* ^d^	*katG* ^e^
**4** (INH)	0.073	0.36	1100	1500	725.80 ± 2.36	2016	<1
**8**	44.70	100	150	300	86.64 ± 3.63	<1	<1
**9**	22.40	74.50	150	300	290.62 ± 1.45	4	1
**10**	29.80	74.50	150	300	160.97 ± 3.89	2	<1
**14**	0.86	3.40	260	340	130.90 ± 3.46	38	<1
**15**	10.30	17.20	510	680	305.45 ± 6.11	18	<1
**16**	0.16	0.33	500	>660	249.41 ± 0.56	756	<1
**20**	0.082	1.60	160	240	81.50 ± 7.23	51	<1
**21**	0.16	0.82	160	240	78.23 ± 0.71	95	<1
**22**	0.16	1.60	160	330	80.19 ± 3.81	50	<1
**23**	1.60	4.90	97.60	240	86.45 ± 4.59	18	<1
**24**	0.81	3.25	81.30	160	105.41 ± 6.13	32	<1
**25**	0.32	6.50	3.25	24.40	98.85 ± 1.48	15	4

^a^ Concentration of compounds exhibiting 50% inhibition of mycobacterial growth, ^b^ concentration of compounds exhibiting 99% inhibition of mycobacterial growth, ^c^ in vitro cytotoxicity, CC_50_—concentration required to reduce the cell growth by 50% compared to untreated control, ^d^ Selectivity Index, SI ratio = CC_50_/MIC_99_*Mtb*, and ^e^ Selectivity Index, SI ratio = CC_50_/MIC_99_ Δ*katG.*

**Table 3 pharmaceuticals-13-00465-t003:** Measured lipophilicity and permeations of compounds **4**, **8**–**10**, **14**–**16**, and **20**–**25**.

Compound	Partition/Distribution Coefficient		PAMPA ^a^
	log *P*	log *D*_7.4_	log *P*_e_
**4** (INH)	−1.00 ± 0.09	-	−6.21 ± 0.10
**8**	0.67 ± 0.07	-	−4.98 ± 0.12
**9**	1.11 ± 0.11	-	−5.09 ± 0.07
**10**	0.99 ± 0.07	-	−4.65 ± 0.09
**14**	2.03 ± 0.09	-	−4.06 ± 0.09
**15**	2.28 ± 0.20	-	−3.90 ± 0.17
**16**	-	0.67 ± 0.04	−5.05 ± 0.30
**20**	1.26 ± 0.22	-	−5.05 ± 0.09
**21**	1.66 ± 0.18	-	−4.40 ± 0.10
**22**	1.30 ± 0.07	-	−4.13 ± 0.07
**23**	0.64 ± 0.09	-	−4.16 ± 0.05
**24**	0.66 ± 0.08	-	−4.85 ± 0.06
**25**	1.15 ± 0.14	-	−4.37 ± 0.04

^a^ log *P*_e_ propranolol −4.82 ± 0.03. PAMPA: parallel artificial membrane permeability.

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
