# Peer review of "Novel Isoniazid-Carborane Hybrids Active In Vitro against Mycobacterium tuberculosis"

_pharmaceuticals, 2020, doi:10.3390/ph13120465_

Round 1

Reviewer 1 Report

The manuscript approaches an important topic of novel antituberculotics with quite an original way. However there is several major issues that needs to be addressed.

First of all I would like to point out the lenght of the manuscript. It is quite extensive I believe making it more comprehensive would help it to be more readable.

Abstract and conclusion need to be improved significantly. Abstract is very detailed in terms of chemistry and does provide a reader with the overal content. Conclusion is again too long and it does not provide reader with the most significant findings of the research. As such, the conclusion is just repetition of the results already given. Are there any leads or is there any benefit of using boron cluster conjugates?

What is the rationale behind using carborane for preparing novel antituberculotics? The increase of lipophilicity is mentioned, but there are surely much more covenient ways to increase lipohilicity than using carboranes conjugates.

The results suggest that that the carborane hybrids are not bioactivated by katG and do not form the adduct to be able to inhibit mycobacterial growth. Is there any other proposed mechanism of action? Is InhA a target structure for the novel compounds? If not, is there any other enzyme that is inhibited?

Are there any structure-activity relationships?

The X-ray experiments results are not further discussed? Are the described interaction relevant for ligand-receptor complex formation?

L57: InhA inhibition is not strictly speaking cause of cell death. It is bacteriostatic in most cases.

L69-70: The citation is needed for the statement that INH derivatives with increased lipophilicity are the most potent drug candidates.

L201-204: The results suggest that the product is unstable. How can you be sure, that the results provided in the in vitro experiments were brought about by closo carborane derivative?

L360: Indomethacin-carborane conjugates are supposed to be more water soluble. However, the carboranes are throughout the manuscript discussed as a moiety to increase lipohilicity. Can this be explained?

L366-373: This part of the text discusses the chemical differences between closo and nido states. However, I fail to see the relevance for antimycobacterial activity, which should be discussed in this part of the text.

L380-381 and 403-414: The proposed mechanism of novel compounds is the same as the one of INH. However, the results suggest that it may be different mechanism. This needs to be further discussed.

L384-396: The exact protocol for this experiment is not provided. The results that are outlined here do not take the steric constrains of bioactivation enzymes into consideration.

The permeability values should be provided as a table or a graph.

Why was DMSO and MeOH used in permeability experiments? This could hardly be called biorelevant conditions.

Author Response

Reviewer 1

 The manuscript approaches an important topic of novel antituberculotics with quite an original way. However there is several major issues that needs to be addressed.

We appreciate Reviewer’s positive opinion on our manuscript and his recommendation to publish this work in the Pharmaceuticals after major revision.

 First of all I would like to point out the lenght of the manuscript. It is quite extensive I believe making it more comprehensive would help it to be more readable.

Our manuscript was sent to Pharmaceuticals as an article. Now it contains 20 pages without literature (Introduction - 1 page, Results and Discussion - 9.5 pages, Material and Methods - 8.5 pages). Describing molecules not yet published in the literature, we had accurately described their synthesis and the analyzes performed to confirm their chemical structure. The Result and Discussion contain the necessary information discussing the obtained results. To make the manuscript more comprehensive, we have included 7 Figures and 3 Tables. The rest of the information can be found in Supplementary Materials (89 pages).

Moreover, since the boron clusters still belong to unexplored areas as pharmacophores or modulators, every report on their use in medicinal chemistry and biological activity brings valuable information, for further developments.                                   

Abstract and conclusion need to be improved significantly. Abstract is very detailed in terms of chemistry and does provide a reader with the overal content. Conclusion is again too long and it does not provide reader with the most significant findings of the research. As such, the conclusion is just repetition of the results already given. Are there any leads or is there any benefit of using boron cluster conjugates?

We appreciate the Referee’s comment. The Abstract was improved significantly. Now, it is read:

“Tuberculosis (TB) is a severe infectious disease with high mortality and morbidity. The emergence of drug-resistant TB has increased the challenge to eliminate this disease. Isoniazid (INH) remains the key and effective component in the therapeutic regimen recommended by World Health Organization (WHO). A series of isoniazid-carborane derivatives containing 1,2-dicarba-closo-dodecaborane, 1,7-dicarba-closo-dodecaborane, 1,12-dicarba-closo-dodecaborane, or 7,8-dicarba-nido-undecaborate anion were synthesized for the first time. The compounds were tested in vitro against Mycobacterium tuberculosis (Mtb) H37Rv strain and its mutant (DkatG) defective in the synthesis of catalase-peroxidase (KatG). N'-[(7,8-dicarba-nido-undecaboranyl)methylidene]isonicotinohydrazide (16) showed the highest activity against wild-type Mtb strain. All hybrids could inhibit the growth of the ΔkatG mutant in lower concentrations than INH. N′-[(1,12-dicarba-closo-dodecaboran-1yl)ethyl]isonicotinohydrazide (25) exhibited more than 60-fold increase in activity against Mtb DkatG as compared to INH. This compound was also found to be noncytotoxic up to a concentration four times higher than the MIC99 value.”

The Conclusions were also shortened. Now, it is read:

       “In the search for new antitubercular agents, we developed a method for the synthesis of INH modified with a closo-ortho-/meta-/para-carborane and nido-carborane with good yields. The present approach can provide a new route for generating for the first time INH derivatives with various boron clusters.

                Selected compounds, containing closo-carborane, exhibited significant activity, and one modified with nido-carborane exhibited potent activity, similar to INH, against Mtb in vitro. The presence of carborane cluster, regardless of its structure, contributed significantly to increasing INH-carborane hybrids activity against the DkatG mutant, in comparison to INH. Additionally, low cytotoxicity of the chosen hybrids combined with their activity against the Mtb or the DkatG mutant, make them a hit compounds that can be developed into a lead candidate for further studies.

      The present work demonstrated that the conjugation of the biological active of isoniazid and inorganic boron cluster, with their properties, are useful in drug design, might enable to develop a novel class of hybrids, lead compounds, with potential antimycobacterial activity. Studies on the development of new hybrids of isoniazid containing boron clusters are currently in progress in our laboratory.”

What is the rationale behind using carborane for preparing novel antituberculotics? The increase of lipophilicity is mentioned, but there are surely much more covenient ways to increase lipohilicity than using carboranes conjugates.

The properties of boron clusters that are useful in drug design, in addition to the lipophilicity mentioned in the manuscript, include the following [Ali, F.; Hosmane, N.S.; Zhu, Y. Molecules 2020, 25, 828; doi.10.3390/molecules25040828]:

  • the unique non-covalent interaction ability, including ionic interaction, σ-hole interaction and dihydrogen bond formation (is different from that pure organic molecules);
  • spherical or ellipsoidal geometry and rigid 3D arrangement, these offer versatile platforms for 3D molecular construction;
  • chemical stability and simultaneous susceptibility to functionalization;
  • bioorthogonality, stability in biological environments and decreased susceptibility to metabolism;
  • abiotic origin, foreignness to existing enzymes and a decreased likelihood of triggering the development of drug resistance;
  • high boron contents (important for BNCT);
  • resistance to ionizing radiation, a feature that is important for the design of radiopharmaceutical agents;

The results suggest that that the carborane hybrids are not bioactivated by katG and do not form the adduct to be able to inhibit mycobacterial growth. Is there any other proposed mechanism of action? Is InhA a target structure for the novel compounds? If not, is there any other enzyme that is inhibited?

We believe that some compounds which are more active against the wild-type strain than ΔkatG mutant form adducts with NAD cofactor (16, 21), others, highly active against ΔkatG mutant probably not. We discussed this more broadly in the updated manuscript (please see the response to the comment: L380-381 and 403-414).

Are there any structure-activity relationships?

We have observed that:

  1. Hydrazides 8-10 were the least active against Mtb H37Rv.
  2. Hydrazones 14, 15 bearing closo-carborane were less active against Mtb H37Rv strain than hydrazones 21, 22 bearing also closo-carborane and the longer linker between carborane cluster and INH residue.
  3. Hydrazones 20-22 were more active against the Mtb H37Rv strain than hydrazides 23-25.
  4. Conjugate 16 modified with nido-carborane was the most active against Mtb H37Rv strain than conjugates modified with closo-carborane.
  5. Conjugate 16 active against Mtb H37Rv strain was not active against DkatG mutant.

We have not noticed structure-activity relationships depending on the isomer of the boron cluster against Mtb H37Rv.

The X-ray experiments results are not further discussed? Are the described interaction relevant for ligand-receptor complex formation?

As suggested by Referee we added to the Results and Discussion (L280-287):

“The interactions described could be relevant in the ligand-receptor complex formation. It was found that 1-carba-closo-dodecaborane (CB11H12), a model for metallacarborane inhibitors of HIV protease, interacts with the building blocks (amino acids) of biomolecules by the formation of dihydrogen bonds, using B-H groups of the carborane cluster. Dihydrogen bonds are mainly electrostatic interactions between negatively charged boron-bound hydrogen atoms and positively charged hydrogen atoms of biomolecules. Another type of interaction was found for C-H ··Y hydrogen-bonded complexes. These complexes were less stable [31,32].”

 L57: InhA inhibition is not strictly speaking cause of cell death. It is bacteriostatic in most cases.

We agree with Referee, isoniazid is bactericidal to rapidly dividing mycobacteria, but is bacteriostatic if the mycobacteria are slow-growing [Ahmad, Z.; Klinkenberg, L.G.; Pinn, M.L.; Fraig, M.M.; Peloquin, C.A.; Biskai, W.R.; Nuermberger, E.L.; Grosset, J.K.; Karakousis, P.C.; Biphasic kill curve of isoniazid reveals the presence of drug-tolerant, not drug-resistant Mycobacterium tuberculosis in the guinea pig. J. Infect. Dis. 2009, 200, 1136-1143]. On the other hand, the INH-target, InhA, is essential for the viability of the cell and prolonged exposition to INH will lead to the cell death.

We have modified the sentence indicated by the Referee (L47-48). Now, it is read:

“This adduct inhibits the enzyme enoyl-acyl carrier protein reductase (InhA) of the fatty acid synthase II (FASII) leading to growth inhibition or death of bacilli.”

L69-70: The citation is needed for the statement that INH derivatives with increased lipophilicity are the most potent drug candidates.

We have added reference (L61): Rodrigues, M.O.; Cantos, J.B.; Montes D’Oca, C.R.; Soares, K.L.; Coelho, T.S.; Piovesan, L.A.; Russowsky, D.; da Silva, P.A.; Montes D’Oca, M.G. Synthesis and antimycobacterial activity of isoniazid derivatives from renewable fatty acids. Bioorg. Med. Chem. 2013, 21, 6910-6914. This citation is number 12.

As recommended by the second Reviewer, three new references were added [25-27]. The numbering of the literature references was adjusted adequately. The total number of literature references is 59 now.

L201-204: The results suggest that the product is unstable. How can you be sure, that the results provided in the in vitro experiments were brought about by closo carborane derivative?

Conjugate 16 was obtained from conjugate bearing closo-ortho-carborane. In the biological assays described in the manuscript only conjugate 16 was tested. Its analog bearing closo-ortho-carborane has not been tested. Compound 16 is stable. Other compounds obtained containing closo-ortho-carborane, 8, 20, 23, are stable and not converted spontaneously to nido-carborane derivatives.

L360: Indomethacin-carborane conjugates are supposed to be more water soluble. However, the carboranes are throughout the manuscript discussed as a moiety to increase lipohilicity. Can this be explained?

 Indomethacin-carborane conjugate bearing ortho-carborane isomer exhibited very low water solubility due to the high lipophilicity of the carborane cluster [Neuman, W.; Xu, S.; Sarosi, M.B.; Scholz, M.S.; Crews, B.C.; Ghebreselasie, K.; Banrjee, S.; Marnett, L.J.; Hey-Hawkins, E. nido-Dicarbaborate induces potent and selective inhibition of cyclooxygenase-2. ChemMedChem 2016, 11, 175-178]. Introduction of the opened form of carborane cluster - nido-carborane led to high water solubility derivative containing amphiphilic nido-carborane.

Hydrophobicity of carborane increases in the order ortho- < meta- < para- [Leśnikowski, Z.J. Challenges and opportunities for the application of boron clusters in drug design. J. Med. Chem. 2016, 59, 7738–7758]. The hydrophobic closo-carborane cages can be transformed, using bases (e.g. fluoride, alkoxides or amines), into amphiphilic anionic clusters with one of their vertices missing. These species are referred to as nido-carboranes [Leśnikowski, Z.J. Recent developments with boron as a platform for novel drug design. Expert Opin. Drug Dis. 2016, 11, 569–578]. Utilizing of a single synthetic step, the binding characteristics of a carborane-containing molecule can be altered from hydrophobic to hydrophilic. These features allow using carboranes to modulate biomolecules’ physicochemical and biological properties.

L366-373: This part of the text discusses the chemical differences between closo and nido states. However, I fail to see the relevance for antimycobacterial activity, which should be discussed in this part of the text.

The relevance of nido- status of boron cluster for its NIH biological activity is stressed in lines L347-349 and is shown in Table. 2. The exact reason why the NIH derivative bearing carborane cage negatively charged (nido-) in comparison with its neutral counterpart is not clear at present and requires further studies. We added the following statement at the end of L356-368 section:

“Indeed, though the type of boron cluster and its closo/nido status may affect biological activities of their derivatives, the cause for this phenomenon is not clear at present and requires further study.”

L380-381 and 403-414: The proposed mechanism of novel compounds is the same as the one of INH. However, the results suggest that it may be different mechanism. This needs to be further discussed.

Additionally, we discussed the activity of the tested compounds and added description (L413-430):

“The formation of a covalent adduct with the NAD cofactor and inhibition of FASII might be not the only mechanism of action of the INH-modified compounds. Hydrazone 20 and hydrazides 23-25 are active against wild-type Mtb but their MIC99 are one order of magnitude higher comparing to INH (4) and 16. On the other hand, they are very active (especially conjugate 25) against mutant ΔkatG. So, at least compound 25 doesn’t require formation of adduct with NAD to present the bacteriostatic/bactericidal effect. We cannot exclude that compound 25 can form the adduct within the wild-type strain, when KatG is available, and that INH-derivatives synthesized here, with or without NAD cofactor, affect the same molecular target as INH. However, it is also likely that other essential molecules of Mtb are targeted by this compound. We speculate, that compounds which are effective against wild-type Mtb, but not ΔkatG mutant (16), need activation by formation of a covalent adduct with the NAD cofactor, and affect InhA, however the compounds which are potent against ΔkatG doesn’t need the intracellular activation and targets other essential molecules within Mtb. We are not able to exclude that some compounds, such as 25, can compose INH-NAD adduct in the wild-type Mtb but it presents the bactericidal effect without NAD cofactor as well, affecting the same or other targets. The identification of the alternative targets for INH-derivatives synthetized in this work needs complex study. Research of direct InhA inhibitors, avoiding the activation step, has emerged as a promising strategy to combat the global spread of MDR-TB [50].

L384-396: The exact protocol for this experiment is not provided. The results that are outlined here do not take the steric constrains of bioactivation enzymes into consideration.

The general procedure for competitive trapping, applied to compounds INH (4), 8, 21, and 25 was described in Supplementary Materials (page 82). MS spectra related to this experiment are also in Supplementary Materials (Figures S80-82).

We would like to strongly emphasize that the carried out experiment may only suggest explanation the of diverse activity of the tested compounds. Pro-drug isoniazid is activated in bacilli by catalase-peroxidase (KatG) to form the INH-nicotinamide adenine dinucleotide (NAD) adduct. In the manuscript, is written: "It is previously reported that access to the heme active site of KatG poses steric constraints, and therefore, any factor influencing the stereochemistry of tested compounds should be relevant [42]."

Additionally, we have corrected the text to make it clear and unambiguous. Now, it is read (L370-395):

"Conjugates 14, 20, and 22-25 exhibited significant activity against Mtb (MIC99 1.6-6.5 µM) but showed lower activity than compounds 16 and 21. Schiff bases 21 and 22 with a longer linker between carborane and INH residue were more active than the appropriate Schiff bases 14 and 15 with a shorter linker.

The different antimycobacterial activity of compounds, 8-10, 14-16, and 20-25 is most probably related to the steric characteristic of the carborane cluster and its impact on the promotion of the formation of the isonicotinoyl radical to form INH-nicotinamide adenine (NAD) adducts.

It is previously reported that access to the heme active site of KatG poses steric constraints, and therefore, any factor influencing the stereochemistry of tested compounds should be relevant [42]. The manganese catalyst [Mniv-Mniv(µ-O)3L2](PF6)2, (L=1,4,7-trimethyl-1,4,7-triazacyclononane) was used, as it was hypothesized to be a mimic for oxidation by the KatG enzyme due to its catalase activity [43]. This catalyst forms a MnV=O species due to its catalase activity [44,45]. When INH was reacted with various oxidant, including manganese catalyst, in the presence of the stable radical trap 2,2,6,6-tetramethylpiperidine-1-oxyl (TEMPO) O-isonicotinoyl hydroxylamine was formed [44,45]. This product is the consequence of acyl radical generation and trapping, as supported by Braslau [46].

In our investigation, INH (4) was reacted in the presence of TEMPO and Mn catalyst, under nitrogen atmosphere in MeOH/ACN (1:99, v/v) with periodic acid as a co-oxidant [47,48]. O-Isonicotinoyl hydroxylamine (26) was formed (Figures. S78, S79, general experimental details, (SM)), and unreacted substrate INH (4) was not observed in the reaction mixture. When elected hybrids 8, 21, and 25, with diverse antimycobacterial activity against Mtb (Table 2), were reacted with a manganese-containing oxidant in the presence of TEMPO, O-isonicotinoyl hydroxylamine (26) was produced. The unreacted substrates were also observed in the reaction mixture (3-46%), for the lowest value for the most active hybrid 21 of these three compounds (Figures S80-S82 (SM)). These results could shed light on the different antimycobacterial of the tested conjugates compared to unmodified INH (4)."

 The permeability values should be provided as a table or a graph.

The permeability values are provided in a table. Please see Table 3.

 Why was DMSO and MeOH used in permeability experiments? This could hardly be called biorelevant conditions.

Parallel artificial membrane permeability (PAMPA) is a relatively high throughput and is important in the early stages of the drug discovery process. This is in vitro technique. As we wrote in the manuscript, the technique involves a donor and an acceptor compartment separated by a filter supporting a liquid artificial membrane. The compound to be tested is placed in the donor compartment and is allowed to permeate between the donor and the acceptor compartments through the artificial membrane. DMSO and MeOH (at a defined concentration) are most commonly used to dissolve the tested compounds. Other assays to determine the permeability using Caco-2, the immobilized artificial membrane (IAM) technique, or immobilized liposome chromatography (ILC) also require the dissolution of tested compounds.

Closing remark: The text of the manuscript was edited for proper English language at Translmed Publishing Group (TPG) (Dallas/Ft. Worth area, 713 Sleepy Hollow Dr., Cedar Hill, TX 75104 U.S.A.)

Reviewer 2 Report

The manuscript by Daria Rozycka et al. describes synthesis and in vitro study of isoniazid-carborane hybrids, some of which demonstrate high activity against Mycobacterium tuberculosis wild-type H37Rv strain and its mutant. The synthesis and characterization of all new compounds and their physicochemical study are well done, the biological studies look fine as well. Therefore, I believe that the manuscript can be published in Pharmaceuticals after minor revision (see below).

It is very advisable to give the formula of isoniazid in the introduction and to use its abbreviation (INH) instead of number of the compound (4) in all tables and figures in order to facilitate its comparison with boron-containing analogues.

The spontaneous self-degradation of different nitrogen-containing derivatives of ortho-carborane is well known (E.Svantesson et al., Acta Chem. Scand., 1999, L.O.Kononov et al., J. Organomet. Chem., 2005, 690, 2769; J. Zhao et al., Inorg. Chem. Commun., 2011, 14, 934) and the corresponding references should be included in the manuscript.

The explanation of the different antimycobacterial activity of the obtained compounds (lines 378-398) is not very clear and requires more clarity.

Author Response

Reviewer 2

The manuscript by Daria Rozycka et al. describes synthesis and in vitro study of isoniazid-carborane hybrids, some of which demonstrate high activity against Mycobacterium tuberculosis wild-type H37Rv strain and its mutant. The synthesis and characterization of all new compounds and their physicochemical study are well done, the biological studies look fine as well. Therefore, I believe that the manuscript can be published in Pharmaceuticals after minor revision (see below).

 We appreciate the Reviewer’s positive opinion on our contribution and his recommendation to publish this work in the Pharmaceuticals after minor revision.

 It is very advisable to give the formula of isoniazid in the introduction and to use its abbreviation (INH) instead of number of the compound (4) in all tables and figures in order to facilitate its comparison with boron-containing analogues.

In the Introduction chemical name of isoniazid: isonicotinic acid-derivative hydrazide (pyridine-4-carbohydrazide) (INH) was given (line 45). The formula of INH is shown in Figure 3 presenting the chemical syntheses of INH derivatives. To fulfill the reviewer's requirements the isoniazid abbreviation, INH was added to the number of the compound (4) in Figure 3, Table 2, Figure 6, Figure 7, Table 3, and in the text. All changes were highlighted, using the "Track Changes" function in Microsoft Word, so that changes are easily visible to the editor and reviewer.  

The spontaneous self-degradation of different nitrogen-containing derivatives of ortho-carborane is well known (E.Svantesson et al., Acta Chem. Scand., 1999, L.O.Kononov et al., J. Organomet. Chem., 2005, 690, 2769; J. Zhao et al., Inorg. Chem. Commun., 2011, 14, 934) and the corresponding references should be included in the manuscript.

We appreciate the Referee’s remark. Additional information related to the transformation of closo-carborane to nido-carborane, in solution, was added (L200-206):

“It was also found that the closo-carborane derivatives can be cleaved to nido-structure in solution. The spontaneous degradation of racemic ortho-carboranylalanine to the corresponding diastereomeric, racemic pairs of nido-carboranylalanine was observed in water-methanol solution [25]. 1,2-Dicarba-closo-dodecaborane-lactose conjugate, dissolved in water or methanol, was easy deboronated to nido-counterpart [26]. The same degradation was observed for disubstituted ortho-carborane, 1,2-bis(aminomethyl)-1,2-dicarba-closo-dodecaborane hydrochloride, in deuterated DMSO [27].”

Three new references were added. The numbering of the literature references was adjusted adequately. As recommended by the first Reviewer, four new references were added ([12, 31, 32, 50]). The total number of literature references is 59 now.

 The explanation of the different antimycobacterial activity of the obtained compounds (lines 378-398) is not very clear and requires more clarity.

We appreciate the Reviewer’s corrections. We have corrected the text to make it clear and unambiguous. However, we would like to strongly emphasize that the carried out experiment may only suggest explaining the diverse activity of the tested compounds.

Now, it is read (L370-395):

"Conjugates 14, 20, and 22-25 exhibited significant activity against Mtb (MIC99 1.6-6.5 µM) but showed lower activity than compounds 16 and 21. Schiff bases 21 and 22 with a longer linker between carborane and INH residue were more active than the appropriate Schiff bases 14 and 15 with a shorter linker.

The different antimycobacterial activity of compounds, 8-10, 14-16, and 20-25 is most probably related to the steric characteristic of the carborane cluster and its impact on the promotion of the formation of the isonicotinoyl radical to form INH-nicotinamide adenine (NAD) adducts.

It is previously reported that access to the heme active site of KatG poses steric constraints, and therefore, any factor influencing the stereochemistry of tested compounds should be relevant [42]. The manganese catalyst [Mniv-Mniv(µ-O)3L2](PF6)2, (L=1,4,7-trimethyl-1,4,7-triazacyclononane) was used, as it was hypothesized to be a mimic for oxidation by the KatG enzyme due to its catalase activity [43]. This catalyst forms a MnV=O species due to its catalase activity [44,45]. When INH was reacted with various oxidant, including manganese catalyst, in the presence of the stable radical trap 2,2,6,6-tetramethylpiperidine-1-oxyl (TEMPO) O-isonicotinoyl hydroxylamine was formed [44,45]. This product is the consequence of acyl radical generation and trapping, as supported by Braslau [46].

In our investigation, INH (4) was reacted in the presence of TEMPO and Mn catalyst, under nitrogen atmosphere in MeOH/ACN (1:99, v/v) with periodic acid as a co-oxidant [47,48]. O-Isonicotinoyl hydroxylamine (26) was formed (Figures. S78, S79, general experimental details, (SM)), and unreacted substrate INH (4) was not observed in the reaction mixture. When elected hybrids 8, 21, and 25, with diverse antimycobacterial activity against Mtb (Table 2), were reacted with a manganese-containing oxidant in the presence of TEMPO, O-isonicotinoyl hydroxylamine (26) was produced. The unreacted substrates were also observed in the reaction mixture (3-46%), for the lowest value for the most active hybrid 21 of these three compounds (Figures S80-S82 (SM)). These results could shed light on the different antimycobacterial of the tested conjugates compared to unmodified INH (4)."

Round 2

Reviewer 1 Report

The manuscript has been significantly improved. I have only minor issue concerning the reference no 12, where authors state, that INH derivatives with greater lipophilicity are emerging as one of the most potential antimycobacterial agents, while citing the work from 2013, that is not really emerging. I appreciate that an effort to propose the mechanism of action has been made. I can only hope the actual mechanism of action for will be investigated in future work for the most promising compounds.